

**Low CO₂ evasion rate from the mangrove surrounding waters of Sundarban**
Anirban Akhand[1], Abhra Chanda[2], Kenta Watanabe[1], Sourav Das[2], Tatsuki Tokoro[1], Kunal
Chakraborty[3], Sugata Hazra[2], Tomohiro Kuwae[1]
[1]Coastal and Estuarine Environment Research Group, Port and Airport Research Institute, 3-1-1
Nagase, Yokosuka 239-0826, Japan.
[2]School of Oceanographic Studies, Jadavpur University, 188, Raja S. C. Mullick Road, Kolkata
700 032, West Bengal, India.
[3]Indian National Centre for Ocean Information Services, Ocean Valley, Pragathi Nagar (BO),
Nizampet (SO), Hyderabad 500 090, India
Running Head: Low CO₂ evasion from mangrove water





**Abstract**
Globally, water bodies adjacent to mangroves are considered sources of atmospheric $CO_2$. We
directly measured the partial pressure of $CO_2$ in water, $pCO_2$(water), and other related
biogeochemical parameters with very high (1-min) temporal resolution at Dhanchi Island in
India's Sundarbans during the post-monsoon season. We used elemental, stable isotopic, and
optical signatures to investigate the sources of dissolved inorganic carbon (DIC) and organic
matter (OM) in these waters. Diel mean $pCO_2$(water) was marginally oversaturated in creeks
(efflux, $69 \pm 180$ µmol m$^{-2}$ h$^{-1}$) and undersaturated along the island boundary and in the main
river (influx, $-17 \pm 53$ and $-31 \pm 73$ µmol m$^{-2}$ h$^{-1}$, respectively) compared to the atmospheric
$CO_2$ concentration. The possibility in earlier studies of over- or underestimating the $CO_2$ flux
because of an inability to capture tidal minima and maxima was minimized in the present study,
which confirmed that the waters surrounding mangroves in this region can act as a sink or a very
weak source of atmospheric $CO_2$. $\delta^{13}C$ values for DIC suggest a mixed DIC source, and a three-
end-member stable isotope mixing model and optical signatures of OM suggest negligible
riverine contribution of freshwater to OM. We conclude that the $CO_2$ sink or weak source
character was due to a reduced input of riverine freshwater [which usually has high $pCO_2$(water)]
and the predominance of $pCO_2$-lean water from the coastal sea, which eventually increases the
buffering capacity of the water as evidenced by the Revelle factor. Up-scaling the $CO_2$ flux data
for all seasons and the entire estuary, we propose that the $CO_2$ evasion rate observed in this study
is much lower than the recently estimated world average. Mangrove areas having such low
emissions should be given due emphasis when up-scaling the global mangrove carbon budget
from regional observations.





Keywords: Air–water $CO_2$ flux; estuary; mangrove-associated water; net $CO_2$ sink; $pCO_2$(water);
stable isotope
**1. Introduction**
According to the Fifth Assessment Report (AR5) of the Intergovernmental Panel on
Climate Change (IPCC) (IPCC, 2014), greenhouse gas emissions increased at a rate of 2.2% per
year during the last decade (2000–2010) and the emission rate reached $49 \pm 4.5$ Pg $CO_2$-
equivalents $y^{-1}$ in 2010. The carbon stocks within several coastal ecosystems (mangroves, tidal
marshes, seagrass beds), collectively referred to as "blue carbon", have drawn attention in this
regard (Donato et al., 2011; McLeod et al., 2011; Pendleton et al., 2012), and initiatives to
characterize the functioning and long-term future of these blue-carbon ecosystems have also
begun (Macreadie et al., 2019). All of these ecosystems are known to be carbon sinks;
mangroves, however, deserve special mention owing to their large soil organic pool and their
being an active deposition centre for both autochthonous and allochthonous organic matter
(Breithaupt et al., 2012; Sanders et al., 2016a, 2016b).
Globally, mangroves reportedly have an estimated mean net primary production of $218 \pm$
72 Tg C $y^{-1}$ (Bouillon et al., 2008; Fuentes and Barr, 2015). Moreover, the combined carbon (C)
stock in mangroves (above ground and live belowground; 956 Mg C $ha^{-1}$) is much higher than
that in other carbon rich ecosystems such as salt marshes (593 Mg C $ha^{-1}$), seagrasses (142 Mg C
$ha^{-1}$), peatland (408 Mg C $ha^{-1}$), or even rain forests (241 Mg C $ha^{-1}$) (Alongi, 2014; Donato et
al., 2011; Twilley et al., 1992). Despite covering only 0.1% of the earth's total land area
(Jennerjahn and Ittekkot, 2002), 0.7% of tropical forests globally (Giri et al., 2011), and 0.5% of





the global coastal ocean (Rosentreter et al., 2018), mangroves are one of the most productive
ecosystems in the world, with high carbon-sequestration potential.

Although mangrove ecosystems as a whole are net sinks for $CO_2$, the waters adjacent to

mangroves (as well as sediments) emit substantial amounts of $CO_2$, because they have substantial
organic carbon loading, which is mainly attributed to mangrove biomass, terrestrial detritus,
microphytobenthos, land-driven allochthonous nutrients, and phytoplankton (Borges et al., 2005,
2018; Borges and Abril, 2011; Bouillon and Boschker, 2006; Kristensen et al., 2008; Lekphet et
al., 2005). In contrast to processes in other forests, tidal flow allows mangroves to exchange both
inorganic and organic solutes and particulates with adjacent water bodies (Adame and Lovelock,
2011). Several studies have attributed the high $CO_2$ emissions from the waters surrounding
mangroves to the efficient exchange and mixing of surface water with pore water through tidal
pumping, leading to enrichment of the partial pressure of $CO_2$ in water [$pCO_2$(water)] and
dissolved inorganic carbon (DIC), as well as metabolic activity in sediments (Bouillon et al.,
2007b; Gleeson et al., 2013; Li et al., 2009; Maher et al., 2013; Robinson et al., 2007; Santos et
al., 2012; Sippo et al., 2016).

Processes such as the mineralization of dissolved organic carbon (DOC) and particulate

organic carbon (POC) lead to additional DIC in water bodies adjacent to mangroves (Gattuso et
al., 1998; Maher et al., 2013, 2015). The mineralization of organic carbon in mangrove
sediments is facilitated through several pathways such as sulfate reduction, iron reduction,
aerobic respiration, and carbonate dissolution (Borges et al., 2003; Kristensen and Alongi, 2006;
Krumins et al., 2013).



Although the mangroves surrounding waters usually act as a source of $CO_2$, there are still

uncertainties with respect to the global budget of these emissions. In their global estimates,
studies carried out during the last decade mostly considered the total mangrove area of the globe
to be the total area of the water adjoining the mangroves (Borges et al., 2003; Koné and Borges,
2008), which is certainly not the actual case (Sippo et al., 2016). Moreover, these studies never
reported negative fluxes (i.e., the water surrounding mangroves acting as a sink for $CO_2$).
However, some recent measurements in mangroves surrounding waters show negative fluxes in
some regions, and at certain times of the year $CO_2$ fluxes in mangrove waters can be negative as
well (Biswas et al., 2004; Akhand et al., 2013b, 2016). In addition, Call et al. (2015) recently
observed very low effluxes (almost zero) in a sub-tropical mangrove ecosystem in Australia. In
this regard, Rosentreter et al. (2018) emphasized the temporal resolution of sampling, which can
lead to considerable uncertainty. They argued that data acquisition at hourly or greater intervals
often misses the tidal maxima and minima, and deducing the mean $CO_2$ flux from such data
might lead to under- or overestimation of fluxes.

At present, all rivers in the Indian part of the Sundarbans except for the Hooghly and its

distributary the Muriganga – namely the Saptamukhi, Thakuran, Matla, Gosaba, and Bidya rivers
– have lost their connections with the main flow of the River Ganga because of siltation in the
upper reaches; their estuarine character is now maintained only by monsoonal runoff (Cole and
Vaidyaraman, 1966; Sarkar et al., 2004). Thus, the central part of the Sundarbans lacks riverine
freshwater input (Chakrabarti, 1998; Mitra et al., 2009).

In the last decade, the air–water $CO_2$ flux has been well studied in the Indian section of

the Sundarbans from the perspective of spatial variability (Akhand et al., 2013b, 2016; Dutta et
al., 2019) and seasonal variability (Biswas et al., 2004; Akhand et al., 2016). Hence, in this study



we do not emphasize these two aspects. Rather, we focus on a particular site and season
previously reported as being a maximum sink or reduced source of $CO_2$ (Biswas et al., 2004;
Akhand et al., 2016) in order to carry out a thorough and comprehensive investigation into the
reasons for such a sink or reduced source. However, as there is no well-defined demarcation of
"mangroves surrounding waters", we set up sampling stations in three types of water bodies
around the mangroves in the study area, namely creeks, island boundaries, and the main river,
which are usually considered as the waters surrounding mangroves. These stations are mostly in
the sub-tidal zone, although the island boundary stations were chosen as close to the intertidal
zone as possible.
We hypothesized that despite being an ecosystem of mangroves surrounded by water, the
$CO_2$ sink or weak source character of the Indian Sundarbans is mainly caused by the negligible
riverine freshwater contribution of carbon species. To examine this hypothesis, we used
automated data collection to obtain high-temporal-resolution data for $pCO_2(water)$ and other
related biogeochemical parameters. To our knowledge, this is the first time in the Sundarban that
this type of data has been collected at this temporal resolution. In addition, we used elemental,
stable isotopic, and optical signatures to quantify the carbon load in the surface water and to
identify the carbon sources at our site, which identified as a $CO_2$ sink or weak source in previous
studies. Finally, we up-scaled our $CO_2$ flux data for all seasons and for the whole estuary and
then compared them with the global average data. Out of the many estuaries that flow through
Sundarbans, we considered the Matla Estuary for up-scaling, as this estuary flows through the
central part of Indian Sundarban and flows for a long distance covering almost the entire north-
south extent of Sundarbans. Moreover, this estuary has been exhaustively studied in the recent
past from the perspective of air–water $CO_2$ flux. Though the present study site is located near the



Thakuran Estuary, we assume that since Matla and Thakuran are adjacent estuaries, there is no
significant difference in their biogeochemical characteristics.
Our objectives were (a) to reduce the uncertainty of flux estimation by direct and
continuous measurement of $pCO_2$(water) and related biogeochemical parameters, (b) to examine
the role of mangrove-derived carbon (DIC, DOC and POC) in the air–water $CO_2$ flux, (c) to
estimate the net ecosystem productivity and calcification rates in the waters around the
mangroves by using high-temporal-resolution biogeochemical data, and finally (d) to identify an
understandable reason behind the transient sink character of this region despite its being
surrounded by dense mangrove vegetation.

**2. Materials and methods**
**2.1 Study area**
The Sundarbans, a UNESCO world heritage site, is the world's largest mangrove forest.
It is situated in the lower stretch of Ganges-Brahmaputra-Meghna (GBM) Delta and extends into
the countries of both India (40%) and Bangladesh (60%) and faces the Bay of Bengal (BoB) to
the south. The present study was carried out in the Indian part of the Sundarbans, which
comprises an area of 10,200 km$^2$ out of which 4200 km$^2$ is demarcated as reserve forest (Ray et
al., 2015). Sampling was conducted in a north–south channel approximately 9 km long (width
varying between 0.70 and 0.85 km) to the west of Dhanchi Island, at the edge of the island, and
in two very narrow (20 m wide) creeks flowing through the island (Fig. 1). For details about the
study area see Section S1 in the supplementary material. Dhanchi Island covers an area of about



33 km$^2$, and its southern tip ends at the BoB. To the east of the island flows the Thankuran River
(7 km wide).

**2.2 Sampling strategy**

The sampling stations were selected in order to understand, characterize and differentiate

the carbon dynamics in three types of waters: waters likely to be strongly influenced by
mangroves, waters under the combined influence of both mangrove-driven pore water and
seawater coming through the channel, and waters minimally influenced by mangroves (Fig. 1).
We selected eight sampling locations (all subtidal) in the westward channel adjoining Dhanchi
Island and in two creeks connected to that channel. One of the creek stations was 9 km north of
the BoB coast (hereafter referred to as C1) and the other 5 km north (C2). We selected three
locations covering the latitudinal extent of the north–south channel very close to the edge of the
Dhanchi Island boundary (10 m from the island's landmass), designated as IB1 (9 km from the
BoB), IB2 (5 km), and IB3 (1 km). The remaining three stations were selected at the same
latitudes as IB1, IB2, and IB3, respectively, only in the middle of the channel, hereafter referred
to as MR1 (9 km from the BoB), MR2 (5 km), and MR3 (1 km). Collectively, the abbreviations
"C", "IB" and "MR" stand for creek, island boundary and mid-river stations, respectively.

**2.3 Sample collection**

Surface pCO$_2$(water), salinity, water temperature, dissolved oxygen, pH, and above-water

photosynthetic photon flux density were monitored with sensors (for details see section 2.4).
Water depth was recorded every hour by using a weighted line and a tape measure. Data for a 24
h diurnal cycle for all of these parameters were acquired at 1 min intervals at each of the eight
stations between 27 January and 6 February 2018. Surface water samples were collected at two





peak low and high tides during the diurnal cycles. In addition, surface water samples were
collected offshore (about 60 km off the coast in the BoB) from on-board a fishing trawler, and
from a station in Diamond Harbour (salinity approx. 0), which served as the marine end member
(MEM) and freshwater end member (FWEM), respectively. The samples were preserved as
described in the next paragraph and sent to the laboratory for analysis.

Samples for total alkalinity (TAlk) and DIC were collected into 250 mL Duran bottles

(SCHOTT AG, Mainz, Germany), filtered through glass-fibre filters (GF/F; Whatman,
Maidstone, Kent, UK) and poisoned with mercuric chloride (200 µL saturated aqueous solution
per bottle) to prevent changes in TAlk and DIC due to biological activity. Samples for DOC and
optical analysis of chromophoric dissolved organic matter (CDOM) were filtered through 0.2 µm
polytetrafluoroethylene (PTFE) filters (DISMIC-25HP; Advantec, Durham, NC, USA) into pre-
combusted (450 °C for 2 h) 100 mL glass vials. DOC samples were acidified with $H_3PO_4$ to a pH
of <2 and then frozen at –20 °C until analysis. CDOM samples were kept at 4 °C until analysis.
Samples for analysis of POC, particulate nitrogen (PN), and for stable isotope analysis ($\delta^{13}C_{POC}$
and $\delta^{15}N_{PN}$) were obtained by filtration (approx. 1–2 L) onto pre-combusted (450 °C for 2 h)
glass-fibre filters (GF/F; Whatman) and stored in the dark at –20 °C until analysis.

Mangrove leaves were collected to determine the stable isotope signatures ($\delta^{13}C_{POC}$ and

$\delta^{15}N_{PN}$) of the mangrove as an end member. Leaves of the dominant mangrove species of the
Indian Sundarbans (Gopal and Chauhan, 2006) – *Avicennia marina*, *Bruguiera gymnorrhiza*,
*Excoecaria agallocha*, and *Phoenix paludosa* – were collected and stored at –20 °C until
analysis.
**2.4 Analytical protocol**





TAlk and DIC concentrations were determined on a batch-sample analyser (ATT-05;
Kimoto Electric Co., Ltd., Osaka, Japan) implementing the Gran Plot method (Dickson et al.,
2007). The accuracy of TAlk and DIC was 3 μmol kg$^{-1}$ water and 4 μmol kg$^{-1}$ water,
respectively, which was confirmed by triplicate measurements of the certified reference material
(CRM) for TAlk and DIC (Kanso Company Ltd., Japan). The pCO$_2$(water) was measured with a
CO$_2$ analyser (Non Dispersive Infrared Sensor) through an equilibrator system (CO$_2$-09, Kimoto
Electric Co., Ltd.) (Kayanne et al., 1995; Tokoro et al., 2014) using a gas-permeable membrane
(Saito et al., 1995). The instrument was calibrated every day at the beginning of the
measurements using pure N$_2$ gas (0 ppm) and span gas (600 ppm CO$_2$ gas with a N$_2$ base;
Chemtron Science Laboratories, India). pH was measured with a sensor (SP-11; Kimoto
Electric). We collected about 10 water samples from different stations and at different times
during sampling and calibrated the pH sensor. The pH of the water samples was calculated from
the measured TAlk and DIC using equilibrium calculations. The pH was finally corrected by the
calibration line from the least-squares regression between the raw sensor data (electromotive
force of the pH electrode) and the difference between the calculated pH and in situ pH, which are
proportional, as indicated by the Nernst equation. The precision of pH and pCO$_2$ was estimated
to be 0.002 and 2 μatm, respectively.
Salinity and temperature were measured with a CT-sensor (INFINITY-CT;
JFE Advantech, Nishinomiya, Japan). Dissolved oxygen (DO) concentration was measured at
hourly intervals with a portable DO meter (FiveGo Series; Mettler Toledo, Germany). DOC
concentration was determined via high-temperature catalytic oxidation with a TOC analyser
(TOC5000A; Shimadzu, Kyoto, Japan). DOM absorbance spectra were recorded from 250 to
700 nm at 1 nm increments using a UV-visible spectrometer (UV-2450; Shimadzu) fitted with a



1 cm quartz flow-cell and referenced to ultrapure water (Milli-Q water; Millipore, Billerica, MA,
USA). The absorbance values at each wavelength were transformed into absorption coefficients
($a_{CDOM}$) by using the following equation:

$$a_{CDOM(\lambda)} = 2.303 \times A_{CDOM(\lambda)} \, (m^{-1}) \, , \qquad\qquad (1)$$


where $A_{CDOM}$ is the absorbance value per metre. The absorption value at 375 nm, $a_{CDOM(375)}$, was
chosen to quantify CDOM because this wavelength has been commonly used in previous studies
to measure DOC absorbance. We also measured $a_{CDOM(254)}$ as a metric of the aromaticity of the
DOM. We calculated specific UV absorption at 254 nm ($SUVA_{254}$, L mg$^{-1}$ m$^{-1}$) as follows:

$$SUVA_{254} = A_{CDOM(254)}/[DOC] \, , \qquad\qquad (2)$$


where [DOC] denotes DOC concentration. Spectral slopes for the interval of 275–295 nm ($S_{275-}$
$_{295}$) were calculated by linear regression of the log-transformed $a_{CDOM}$ spectra. Slopes are
reported as positive numbers to follow the mathematical convention of fitting to an exponential
decay.

Samples for analysis of POC and PN content and stable isotope signatures ($\delta^{13}$C and

$\delta^{15}$N) were dried in an oven at 60 °C. To remove inorganic carbon, we acidified the samples with
1 N HCl and dried them again at 60 °C. POC and PN concentrations and stable isotope





signatures were measured with an isotope-ratio mass spectrometer (Delta Plus Advantage;
Thermo Electron, Bremen, Germany) coupled with an elemental analyser (Flash EA 1112;
Thermo Electron). Stable isotope ratios are expressed in δ notation as the deviation from
standards in parts per thousand (‰) according to the following equation:

$\delta_{13}C, \delta_{15}N = [R_{sample}/R_{standard} - 1] \times 10^3$ , (3)

where R is $^{13}C/^{12}C$ or $^{15}N/^{14}N$. Vienna PeeDee Belemnite (VPDB) and atmospheric nitrogen
were used as the isotope standards for carbon and nitrogen, respectively. The analytical precision
of the Delta Plus Advantage mass-spectrometer system, based on the standard deviation of the
internal reference replicates, was <0.2‰ for both $\delta^{13}C$ and $\delta^{15}N$. The $\delta^{13}C_{DIC}$ was also measured
with the same isotope-ratio mass spectrometer following the method of Miyajima et al. (1995).
**2.5 Estimation of air–water CO$_2$ flux, net ecosystem production (NEP), and net ecosystem**
**calcification (NEC)**
The air–water CO$_2$ flux ($F_{CO2}$, µmol CO$_2$ m$^{-2}$ h$^{-1}$) was determined by the following
equation:

$F_{CO2} = k \cdot K_0 \cdot \Delta fCO_2$ , (4)





where $k$ is the gas transfer velocity (cm h$^{-1}$), K$_0$ denotes the solubility coefficient of CO$_2$ (mol m$^-$
$^3$ atm$^{-1}$), and $\Delta f$CO$_2$ denotes the difference in fugacity ($\approx$ partial pressure) of CO$_2$ between water
and air [$f$CO$_{2(water)}$ – $f$CO$_{2(air)}$]. A positive F$_{CO2}$ value indicates CO$_2$ efflux from the water to the
atmosphere and vice versa. The parameter $k$ was calculated according to formula of Ho et al.
(2011) based on wind speed. These formulae were selected for the present study as the estuarine
channels in the Indian part of the Sundarbans are much wider and with little to hinder free-
flowing wind compared to conditions in Hudson Bay observed by Ho et al. (2014). This formula
was obtained by deploying tracers for measuring the gas transfer velocity in the Hudson River:

$k = (a + 0.266\ \mathrm{U}_{10}{}^2)\ (\mathrm{Sc}/600)^{-0.5}$ ,                       (5)


where U$_{10}$ is the wind speed at 10 m height and $a$ is a constant accounting for gas transfer from
bottom-shear-driven turbulence. Wind speed data were acquired by using a handheld
anemometer (AM 4201; Lutron Inc, Singapore.) and corrected for 10 m height (Kondo, 2000).
Sc is the Schmidt number of CO$_2$ as given by Wanninkhof (2014). K$_0$ is computed based on the
equation given by Weiss (1974).

Net ecosystem production (NEP) and net ecosystem calcification (NEC) were computed

from estimated DIC and TAlk (for details see Section S2 in the supplementary material).
**2.6 Data analysis**

To estimate the biological or physicochemical formation, transformation, and

consumption of carbon species in the water column, we calculated the difference between the



observed concentration of an element ($X$) and its concentration predicted by conservative mixing
($X_{mix}$) as $\Delta X$ for each station: $\Delta X = X - X_{mix}$. The $\Delta X$ concentrations were determined for TAlk,
DIC, DOC, $\delta^{13}C_{DIC}$, $a_{CDOM(375)}$, $a_{CDOM(254)}$, SUVA$_{254}$, and S$_{275-295}$. Predicted conservative
concentrations ($X_{mix}$) were estimated by using a linear salinity mixing model for TAlk, DIC,
DOC, $a_{CDOM(375)}$, and $a_{CDOM(254)}$. Predicted conservative mixing lines (nonlinear) for SUVA$_{254}$
and S$_{275-295}$ were drawn from the concentrations of previously mentioned parameters, whereas
$X_{mix}$ of the $\delta^{13}C_{DIC}$ were adopted according to the formula proposed by Mook and Tan (1991).

The Hooghly River is the main "artery" (Ray et al., 2018a) and only possible source of

riverine freshwater for the Indian Sundarbans, and the adjacent sea is the BoB. For this reason,
the near-zero salinity regime of the Hooghly River and the northern BoB have been universally
used in previous works (Ray et al., 2015, 2018a; Ray and Shahraki, 2016; Dutta et al., 2019) as
freshwater and marine end-members, respectively, in their mixing models for the Indian part of
the Sundarbans. As in these previous studies, we defined the observed salinity at the Diamond
Harbour station (salinity approx. 0) as a proxy for the riverine freshwater end member (FWEM)
and that offshore in the BoB (approx. 60 km off the coast) as the marine end member (MEM).

The FWEM and MEM samples were collected in triplicate simultaneously during our

sampling in the Sundarbans. Our measured TAlk (1646 µmol kg$^{-1}$) and DIC (1476 µmol kg$^{-1}$) of
the MEM is in agreement with the post-monsoon data of Akhand et al. (2012, 2013a). In
contrast, Goyet et al. (1999) reported TAlk and total CO$_2$ of 2180 and 1852 µmol kg$^{-1}$,
respectively, in the surface water of the BoB, although their study site was far offshore (i.e. at
around 10° N latitude). Also, TAlk and DIC of the MEM (same site as the present study) were
higher during other seasons: TAlk pre-monsoon, 1932 µmol kg$^{-1}$; monsoon, 1879 µmol kg$^{-1}$;
DIC pre-monsoon, 1744 µmol kg$^{-1}$; monsoon, 1643 µmol kg$^{-1}$; (unpublished data, Akhand et





al.). Hence, it is apparent that both TAlk and DIC are lower in the northern BoB than farther
offshore, and both decrease further during the post-monsoon season. The lower values might be
due to biogenic activity, as the post-monsoon season is when phytoplankton blooms form in this
region. However, published and unpublished data consistently show that the BoB is $pCO_2$-lean
and the TAlk:DIC ratio is >1.

We used the Bayesian isotopic modelling package Stable Isotope Analysis in R (SIAR)

(Parnell et al., 2010) to partition the proportional contributions of potential organic matter (OM)
sources to the bulk particulate organic matter (POM) based on their N/C, $\delta^{13}C$, and $\delta^{15}N$
signatures. The SIAR model works by determining the probability distributions of the sources
that contribute to the observed mixed signal while accounting for the uncertainty in the
signatures of the sources and isotopic fractionation. We assumed an isotopic fractionation of zero
and ran the model through $1 \times 10^6$ iterations. For each potential source, we report the median and
95% credible interval (CI) of the estimated proportional contribution to the observed value. We
defined three sources – freshwater-derived OM, mangrove-plant-derived OM, and marine OM –
as end members for the isotopic and elemental mixing model.

To characterize the pathway of mineralization of organic matter in this study, TAlk and

DIC were both normalized with respect to salinity. We analysed the stoichiometric relationship
(the slope) between salinity-normalized TAlk (nTAlk) and salinity-normalized DIC (nDIC). DIC
was normalized according to the following equation (Friis et al., 2003):

$nDIC = \{[(DIC_{meas} - DIC_{s=0})/S_{meas}] \times S_{mean}\} + DIC_{s=0}$ ,                    (6)




where $DIC_{meas}$ is the measured DIC, $DIC_{s=0}$ is the DIC of the FWEM (i.e. where salinity = 0),
$S_{meas}$ is the measured salinity, and $S_{mean}$ is the mean salinity, which is used for normalization
(25.0 for this study). TAlk was also normalized using the same equation, replacing $DIC_{meas}$ and
$DIC_{s=0}$ with $TAlk_{meas}$ and $TAlk_{s=0}$, respectively.

The Revelle factor indicates the resistance of the ocean surface layer to absorbing

atmospheric $CO_2$. We used $CO_2SYS$ software to estimate the Revelle factor from the measured
TAlk and DIC at the eight C, IB, and MR stations and at the FWEM and MEM stations.

**3. Results**
**3.1 Physicochemical setting**

The average depths of the C, IB, and MR stations were 4, 3, and 6 m, respectively. The

salinity of the FWEM was almost zero (0.39), and the salinity of the MEM was 26.9. All of the
mean values, standard deviations, and ranges of physicochemical, stable isotopic, and carbonate-
chemistry parameters are in Table S1 (supplementary material). The photosynthetic photon flux
at the water surface varied between $770 \pm 517$ µmol m$^{-2}$ s$^{-1}$ (mean ± s.d.; $n = 1406$) and $834 \pm$
$550$ µmol m$^{-2}$ s$^{-1}$ ($n = 1831$) (see Fig. S1 in the supplementary material for the diel variability).
Water temperature varied within the narrow range between $21.18 \pm 0.66$ °C ($n = 2880$) and 21.79
$\pm 0.49$ °C ($n = 4320$) (Fig. 2). The salinity also varied within a very narrow range. The salinities
were $25.37 \pm 0.65$ at C stations ($n = 2880$), $25.63 \pm 0.46$ at IB stations ($n = 4320$), and $25.62 \pm$
$0.35$ at MR stations ($n = 4320$). DO concentrations gradually and slightly increased from C (5.1
$\pm 0.2$ mg L$^{-1}$; $n = 48$), to IB ($5.5 \pm 0.2$ mg L$^{-1}$; $n = 72$) and MR stations ($5.9 \pm 0.2$ mg L$^{-1}$; $n =$

72).



## 3.2 Carbonate-chemistry parameters

The maximum $pCO_2$(water) was at C ($470 \pm 162$ µatm, $n = 2880$), and decreased

gradually toward IB ($393 \pm 48$ µatm, $n = 4320$) and MR ($380 \pm 66$ µatm, $n = 4320$) (Fig. 2).

Similarly, pH increased from $8.023 \pm 0.015$ ($n = 1337$) at C to $8.032 \pm 0.009$ ($n = 720$) at IB

(Table S1). The pH at MR, however, was $8.030 \pm 0.002$ ($n = 1440$), which is almost same as that

at IB. Measured TAlk showed a similar trend: C ($2047 \pm 289$ µmol kg$^{-1}$; $n = 8$), IB ($1936 \pm 146$

µmol kg$^{-1}$; $n = 11$) and MR ($1887 \pm 19$ µmol kg$^{-1}$; $n = 12$). The measured DIC data showed a

gradual decrease from C ($2219 \pm 244$ µmol kg$^{-1}$; $n = 8$), to IB ($2112 \pm 120$ µmol kg$^{-1}$; $n = 11$)

and MR ($2078 \pm 17$ µmol kg$^{-1}$; $n = 12$). The measured TAlk and DIC of the FWEM were 2977

and 2950 µmol kg$^{-1}$, respectively. The measured TAlk and DIC of the MEM were much lower at

1647 and 1476 µmol kg$^{-1}$, respectively).

We plotted all of the TAlk and DIC values from our sampling sites onto the same graph

along with the conservative mixing line (Fig. 3a and 3b). $\delta^{13}$C of DIC varied over a wide range,

from –1.5‰ to –7.6‰ with a mean of –2.4‰ $\pm$ 1.3‰ ($n = 31$). The $\delta^{13}$C values for DIC were –

6.6‰ and –1.2‰ in the FWEM and MEM, respectively. Most of the DIC $\delta^{13}$C values from C

stations plotted below the conservative mixing line, whereas values from IB and MR stations

plotted in a mixed fashion (below, along, or above the conservative mixing line; Fig. 3i). The

mean Revelle factors for the C, IB, and MR stations were $12.8 \pm 2.1$ (range, 11.2–17.9, $n = 8$),

$12.4 \pm 1.4$ (11.5–16.3, $n = 11$), and $11.8 \pm 0.3$ (11.5–12.6, $n = 12$), respectively. The Revelle

factors for the MEM and FWEM were 10.7 and 26.7, respectively.

## 3.3 Air–water CO₂ flux



All three sampling zones (i.e. C, IB, and MR) varied diurnally between acting as a sink or
source of atmospheric $CO_2$ (Fig. 2). The creeks acted as net sources of $CO_2$ with a mean flux of
$69 \pm 180$ µmol m$^{-2}$ h$^{-1}$ ($n = 2880$) and a range of 104–887 µmol m$^{-2}$ h$^{-1}$. In contrast, the island
boundary and mid-river acted as net sinks for $CO_2$ with mean fluxes of $-17 \pm 53$ µmol m$^{-2}$ h$^{-1}$ ($n$
$= 4320$) and $-31 \pm 73$ µmol m$^{-2}$ h$^{-1}$ ($n = 4320$), respectively (Table S1). The flux ranged between
$-108$ µmol m$^{-2}$ h$^{-1}$ and 225 µmol m$^{-2}$ h$^{-1}$ at IB, and between $-99$ µmol m$^{-2}$ h$^{-1}$ and 251 µmol m$^{-2}$
h$^{-1}$ at MR. We observed a distinct switch from a net $CO_2$ source at the creek stations to a net sink
at the island boundary. Moreover, the magnitude of the sink increased from IB to MR.

**3.4 NEP and NEC**

The photosynthesis–irradiance curve-fitting with measured NEP and NEC yielded
statistically significant relationships only at C2 among the eight stations (Fig. 4). NEP and NEC
at C2 were $-239$ mmol m$^{-2}$ d$^{-1}$ ($R^2 = 0.66$, $P < 0.0001$) and $-149$ mmol m$^{-2}$ d$^{-1}$ ($R^2 = 0.66$, $P <$
0.0001) respectively.

**3.5 Organic matter input**

Most of the DOC values were higher than those predicted by conservative mixing (Fig.
3c). The changes in POC, PN, POC:PN, $\delta^{13}C_{POC}$, and $\delta^{15}N_{PN}$ with salinity are shown in Fig. 3d–
3h. The $\delta^{13}C$ of TOC of leaves from the four dominant mangrove species was $-29.0‰ \pm 1.9‰$
(mean $\pm$ SD), and the $\delta^{15}N$ of TN was $3.2‰ \pm 0.8‰$ (see Table S2 in the supplementary material
for details of elemental and stable isotopic signatures of mangrove leaves as end members for the
mixing model). The $\delta^{13}C$ values of POC from the FWEM and MEM were $-23.9‰ \pm 0.1‰$ ($n =$
3) and $-21.8‰ \pm 0.2‰$ ($n = 3$), respectively, whereas the $\delta^{15}N$ of PN were $6.0‰ \pm 0.6‰$ ($n = 3$)
and $4.1‰ \pm 0.2‰$ ($n = 3$), respectively. The isotopic and elemental signatures of organic matter



sources and bulk POM are presented in Fig. S2 (supplementary material). The results of the OM
mixing model using three parameters (including the N:C ratio) and using two parameters
(excluding the N:C ratio) are presented in Tables S3 and S4 (supplementary material).

The optical signatures $a_{CDOM(254)}$, $a_{CDOM(375)}$, and $SUVA_{254}$ showed a gradual decreasing trend

along with increasing salinity. However, no such trend was found for $S_{275-295}$. Most values for
$a_{CDOM(254)}$, $a_{CDOM(375)}$, and $SUVA_{254}$ plotted below the conservative mixing line, indicating that
their values were lower than the predicted conservative mixing values with a few exceptions,
mainly from the creeks (Fig. S3a–3c; supplementary material). In contrast, almost all of the $S_{275-}$
$_{295}$ values were higher than the predicted conservative mixing values with a sole exception from
MR (Fig. S3d, supplementary material).

**4. Discussion**
**4.1 Sinks and low effluxes of CO₂**

Although the narrow (and shallow) mangrove creeks acted as a weak source of $CO_2$, the

island boundary and middle of the river (which are within few metres of the creeks) clearly acted
as a net $CO_2$ sink. We compared the $pCO_2$(water) and air–water $CO_2$ flux with values of other
studies conducted around the world (Table 1). We considered only recent studies that used high-
resolution direct measurement of $pCO_2$(water) or direct measurement of air–water $CO_2$ flux for
this comparison. From the data in Table 1 it is evident that, except for our study, none of the
studies carried out in the waters around mangroves reported a negative flux; i.e. the waters in the
respective study areas did not act as a $CO_2$ sink. Moreover, in terms of the magnitude of the $CO_2$





source, our value for the creeks was much less than the $CO_2$ emission rates reported by other
studies.

If we consider only the mean $CO_2$ emission rate from the creeks (i.e. $1.6 \pm 4.3$ mmol m$^{-2}$

d$^{-1}$), it is still lower by a factor of about 33 than the latest estimate of world average $CO_2$
emissions from the waters around mangroves ($56.8 \pm 8.9$ mmol m$^{-2}$ d$^{-1}$; Rosentreter et al., 2018).
This low $CO_2$ flux value and the sink characteristics that we observed cannot be explained only
by proximity to a marine environment, because in similarly marine locations the waters
surrounding mangroves are a substantial source of $CO_2$.

**4.2 Effect of tidal pumping and pore water on pCO₂**

The waters around mangroves usually exhibit significant diel variability in terms of both

pCO$_2$(water) and air–water $CO_2$ flux, with higher pCO$_2$(water) values during low tide and vice-
versa (Zablocki et al., 2011). However, our continuous high-temporal-resolution measurements
showed that these changes in pCO$_2$(water) with respect to tides occurred only at the creek
stations (Fig. 2). Except for station IB1, none of the island boundary or mid-river stations
showed such distinct diel variability. The high pCO$_2$(water) during low tide is generally
attributed to pCO$_2$-rich pore water as well as groundwater, and the lower pCO$_2$(water) during
flood tide results from the dilution of mangrove-derived water with pCO$_2$-lean seawater (Maher
et al., 2013; Call et al., 2015; Akhand et al., 2016). The absence of pCO$_2$(water) maxima during
low tide at IB and MR stations indicates that the effect of pore water, which is prominent at the
creek stations, did not play a significant role in regulating the diel variation of pCO$_2$(water) at
these stations. This result might be due to the rapid transport of DIC-rich pore water by the





pCO$_2$-lean seawater leading to rapid dilution of DIC at IB and MR stations compared to stations
C1 and C2.

pCO$_2$(water) and DO usually show a negative correlation, as the consumption of

bicarbonate by phytoplankton during photosynthetic activity tends to convert pCO$_2$(water) to
bicarbonate ions [i.e. a decrease in pCO$_2$(water)] and produces oxygen. Because the temporal
resolution of DO measurements was much coarser than that of our pCO$_2$(water) data, we chose
not to perform a correlation analysis. However, it was clear that in the creeks where pCO$_2$(water)
was highest, the DO values were lowest, whereas the reverse was found at the MR stations. The
photosynthetic potential and composition of phytoplankton would not vary widely between these
stations because they are so close to each other. The atmospheric forcing that also plays a key role
in regulating DO in these waters also should not differ. Yet, even though the stations were close
to each other, the DO maxima during the midday hours (i.e. at the time of highest photosynthetic
photon flux) increased steadily from C to IB to MR stations. Hence, the difference in DO could be
attributed solely to the net heterotrophy exhibited by the water around the mangroves in the creeks
and the net autotrophic conditions at IB and MR. Furthermore, the pore water, which has a
prominent effect in the creeks, cannot promote the same level of heterotrophy when the water mass
is diluted by pCO$_2$-lean seawater.
**4.3 Mechanisms for influx or reduced efflux of CO$_2$**

The Revelle factor is the ratio of the relative change of pCO$_2$(water) to the corresponding

relative change of DIC in marine water (Egleston et al., 2010) and thus reflects the carbonate
buffering capacity of the water mass. Globally, the Revelle factor is found to vary between 8 (in
warm waters) and 15 (in cold waters) (Broecker et al., 1979). The lower the Revelle factor the





greater the capacity of the water to take up $CO_2$. In the present study the Revelle factor was
higher in the creeks than at mid-river and island-boundary stations, signifying a lower potential
for $CO_2$ uptake in the creeks than in the IB and MR waters. In other words, the unit increase in
$pCO_2$(water) with respect to the unit input of DIC was much higher at the creek stations than at
the IB and MR stations.
Taillardat et al. (2018) reported that in pore water, the longer residence time of water
coupled with greater water turnover enhances the solute fraction, which in turn increases the
$pCO_2$(water). This phenomenon was found to take place in the creeks, where in the absence of
rapid mixing, the mangrove water mass had a higher residence time compared to the island
boundary and mid-river, where the pore water underwent rapid dilution with $pCO_2$-lean seawater
leading to a reduced Revelle factor. The Revelle factor of the FWEM (26.7) was very high
compared to its values at the C, IB, or MR stations. Although the riverine freshwater supply is
limited in our study area, some ends up in the river mouth. Moreover, the estimated Revelle
factor of the pore water of Sundarban mangroves (computed from pore-water pH data of Mandal
et al. [2009] and pore-water DIC data of Dutta et al. [2019]) was also much higher (17.6) than
the Revelle factor in the creeks. Pore water is generated exclusively within the mangrove
environment.
In addition to pore water and freshwater, estuaries also often experience groundwater
seepage. The Revelle factor of groundwater in and around Sundarban mangroves was $14.9 \pm 1.0$
(unpublished data, Akhand et al.). However, the Revelle factors of riverine freshwater, pore
water and ground water were substantially higher than the Revelle factors at C, IB, and MR.
However, despite the higher Revelle factors for pore water, groundwater, and freshwater, the
waters around mangroves had lower values. The most likely explanation for such lower values is





the influence of marine water. The Revelle factor of the MEM (10.7) was substantially lower
than the factors in creek, island-boundary and mid-river waters. The Revelle factors farther
offshore in the BoB (almost to 10° N latitude) estimated from the TAlk and DIC data of Goyet et
al. (1999) were only 8.9. When waters with such a low Revelle factor are mixed with the
groundwater, pore water, and freshwater at the C, IB, and MR stations, the resultant water
masses have an intermediate buffering capacity, between those of the FWEM and MEM. In other
words, the buffering capacity of the waters adjacent to mangroves is enhanced by mixing with
MEM waters.
Measured DIC in the present study ($2127 \pm 149$ µmol kg$^{-1}$ [mean $\pm$ SD]; range, 2045–
2732 µmol kg$^{-1}$; $n = 31$) was in agreement with the reported DIC ranges from other mangrove-
associated waters that acted as net sources of $CO_2$ (Zablocki et al., 2011; Linto et al., 2014;
David et al., 2018; Ray et al., 2018b). DIC data from this study are comparable to DIC data
reported from other stations in the Indian Sundarbans (Ray et al., 2018a), where $pCO_2$(water)
data indicated that those sites might be sources of $CO_2$. In contrast, Dutta et al. (2019) observed
lower DIC values (1680–1920 µmol kg$^{-1}$) at various sites in the Indian Sundarbans during the
same post-monsoon season. Our plot of $\Delta$TAlk:$\Delta$DIC vs. $\Delta pCO_2$ provides additional useful
insights into regulation of the $pCO_2$(water) concentration and subsequently the air–water $CO_2$
flux (Fig. 5a). This plot shows that, except for a few creek data, the buffering capacity was
mostly higher in the IB and MR waters. Hence, we conclude that the buffering capacity in the
study area is mainly governed by intruding $pCO_2$-lean water from the coastal ocean. The waters
around mangroves are known to have a buffering capacity against coastal acidification (Sippo et
al., 2016; Maher et al., 2018). In the present study, the data for the water around mangroves (and
the pore water or groundwater data from other studies) consistently show a Talk:DIC ratio >1,



which also supports this hypothesis. Moreover, as expected for BoB water, the buffering capacity
of the MEM predominates over the mangrove waters, further enhancing the $CO_2$ sink
characteristics of the waters adjacent to mangroves.

The $\delta^{13}C_{DIC}$ values (Fig. 3i) in the creeks were mostly lighter than the conservative

mixing estimates, a clear indication that the mangrove environment is the source of additional
DIC. The deviation plot ($\Delta DIC$ vs. $\Delta\delta^{13}C_{DIC}$; Fig. 5b) shows a positive $\Delta DIC$ with negative
$\Delta\delta^{13}C_{DIC}$, suggesting OM degradation and ground- and pore-water intrusion as the main sources
of DIC. A few data showed a slightly positive $\Delta DIC$ with positive $\Delta\delta^{13}C_{DIC}$ values, which
indicates carbonate dissolution. The most plausible explanation for the sources of the DIC is a
combination of mineralization of mangrove tissue (–29.0‰ ± 1.9‰, present study), degradation
and/or decomposition of POM in the water around mangroves ($\delta^{13}C_{POC}$ = –22.8‰, [present
study]; –23.3‰ [Dutta et al., 2019]), groundwater- and pore water-derived DIC ($\delta^{13}C_{DIC}$ = –
18.0‰ in Sundarban [Dutta et al., 2019]; –14.5‰ to –10.0‰ [Maher et al., 2013]), and
degradation and/or decomposition of marine phytoplankton ($\delta^{13}C_{POC}$ = –21.8‰ [MEM, present
study]; –22.0‰ to –20.0‰ [Rosentreter et al., 2018]). However, we found no significant
correlation between $\delta^{13}C_{DIC}$ and $\delta^{13}C_{POC}$, which might be because the sources were well-mixed.

This type of mixed source has been found in other mangrove environments such as in

arid Red Sea mangroves (–11.2‰ to –15.9‰; $\delta^{13}C_{DIC}$; Sea et al., 2018) and on the north-eastern
coast of Queensland, Australia (–21.7‰ to –8.9‰; Rosentreter et al., 2018). In contrast to mixed
sources, the DIC of mangrove creek was found typically from the mangrove origin near North
Stradbroke Island, Australia (–29.4‰; Maher et al., 2017). However, Rosentreter et al. (2018)
stated that if, in fact, the emitted $CO_2$ was from an allochthonous carbon source, this would have
major implications for mangrove carbon budgets. In other words, under such a scenario the



carbon dynamics in the mangrove-associated waters cannot be completely explained solely by
mangrove-derived DIC loading, but is also regulated by other allochthonous sources.
Ray et al. (2018a) emphasized the rapid transport of material from Sundarban mangroves
to the BoB compared to other mangroves of the world. They assumed that the reason for this
rapid transport is the shorter residence time due to large tidal amplitudes. Because of the
estuarine geometry common in the Sundarbans, a "funnelling effect" tends to amplify the tide,
resulting in a further amplification of the tidal range and faster material transport (Ray et al.,
2018a). Recently Maher et al. (2018) revealed that the export of DIC and TAlk in a subtropical
mangrove system results in a long-term atmospheric carbon sink that is approximately 1.7 times
higher than burial and should be incorporated into the blue-carbon budget. Our observations
from the present study similarly indicates that the BoB could be a potential sink of the
mangrove-derived DIC and TAlk transported laterally from the Sundarbans.
The diagenesis of organic carbon in mangroves takes place through several anaerobic
pathways that increase pore-water TAlk (Koné and Borges, 2008). The relationship between
salinity-normalized DIC and salinity-normalized alkalinity (slope) can indicate the primary
biogeochemical processes that regulate the DIC and alkalinity dynamics (Borges et al., 2003;
Bouillon et al., 2007b). The salinity-normalized DIC and TAlk at our study site correlated well
($r^2 = 0.98$; $n = 31$), having a slope of 0.84 (Fig. 6). The magnitude of the slope is apparently
close to that suggesting denitrification as observed in the waters around mangroves in the
Mekong Delta, Vietnam (Alongi et al., 2000) and Gaji Bay, Kenya (Bouillon et al., 2007a).
Denitrification has also been reported in the mangrove sediment of the Indian (Das et al., 2013;
Ray et al., 2014) and Bangladesh (Neogi et al., 2016) parts of the Sundarbans, and from the
mangrove environment of Goa (India), where it accounts for <0% to 72% of the pore-water


nitrate reduction (Fernandes et al., 2010, 2012). As the present study is the first of its kind in
Sundarbans, our results can be considered baseline data; no comparison with other parts of the
Sundarbans is possible, but this might be an avenue of future research. Although carbonate
dissolution is also evident from our NEC data, this was not a major pathway as is evident from
the deviation plot ($\Delta$DIC vs. $\Delta\delta^{13}C_{DIC}$) (Fig. 4b) and the plot of salinity-normalized DIC vs.
TAlk (Fig. 6). Globally, denitrification is typically considered a minor diagenetic carbon
degradation pathway, with sulfate reduction and aerobic respiration often being the major
pathways in mangrove sediments (Alongi, 2005; Bouillon et al., 2007b).

The significantly negative NEP values indicate net heterotrophy at C2, which is also in

agreement with the mean $CO_2$ flux data, as this station acted as a net source of $CO_2$. Also, a
significant negative NEC value indicates dissolution of calcium carbonate, as reported previously
in Sundarban (Ghosh et al., 1987) and the Hooghly estuaries (Samanta et al., 2015). Mangrove,
seagrass, and saltmarsh ecosystems are likely sites of net carbonate dissolution (Saderne et al.,
2019). Because the changes in DIC and TAlk at C2 could be explained by the photosynthesis–
irradiance curve, it is likely that biological metabolic activities, such as those NEP and NEC,
play a significant role in regulating the autochthonous DIC and TAlk. However, at the other
stations, the mixing of other water masses (leading to allochthonous input of DIC and TAlk) or
other unknown factors (see Tokoro et al., 2014) might be why the changes in DIC and TAlk
could not be explained by the photosynthesis–irradiance curve. To the best of our knowledge, the
present study is the first to estimate NEP and NEC in the waters around mangroves. Hence, these
are also baseline data and cannot yet be compared with other NEP and NEC data from water
around mangroves. However, Saderne et al. (2019) reported a burial rate for inorganic carbon of
0.8 in a mangrove ecosystem.



**4.4 Role of OM loading**


Although the organic carbon pool is not directly related to the air–water $CO_2$ flux, the
DOC and POC pools are important in discussions regarding the air–water $CO_2$ flux in mangrove
environments. One possible reason that mangrove waters function as a net $CO_2$ source is that the
water column and sediments receive substantial quantities of leaf and wood litter from the
overlying canopy and labile organic carbon is exported from mangroves to adjacent aquatic
systems (Borges et al., 2003). The loss of carbon (as DIC) mainly takes place through
mineralization of both DOC and POC, which are eventually exported into the estuaries and
oceans and finally to the atmosphere as $CO_2$ or $CH_4$ (Gattuso et al., 1998; Maher et al., 2013,

2015).

In the present study, DOC comprised the majority of the total organic carbon pool
(approx. 77%) and it was found to be added from within the mangrove surrounding water.
However, the concentrations of both DOC and POC in the present study are much lower (104.5 ±
18.7 µM and 30.7 ± 11.6 µM, respectively) than those observed at other mangrove sites around
the world. For example, DOC in mangroves along the Iranian coast of the Persian Gulf varies
between 296.2 ± 22.7 µM and 332.1 ± 45.6 µM, whereas POC varies between 185.4 ± 98.4 µM
and 481.5 ± 133 µM (Ray and Weigt, 2018). In the mangrove environment of French Guiana
(South America) the DOC and POC vary from 109 to 808 µM and 16.6 to 101 µM, respectively
(Ray et al., 2018b). In Moreton Bay, Australia, DOC is 80–200 µM and POC is 200–400 µM
(Maher et al., 2013). Although the POC concentrations in the present study are in agreement with
the reported ranges from other Sundarban sites (32.0 ± 8.5 µM [May] and 43.8 ± 8.5 µM
[December], Ray and Sahraki [2016]; 80–436 µM [post-monsoon season], Dutta et al. [2019]),
the DOC concentrations are substantially lower than not only other mangrove systems





worldwide, but also other study sites in the Sundarbans (294.2 ± 34.0 µM [May] and 262.0 ±
43.7 µM [December], Ray and Sahraki [2016]; 154–315 µM [post-monsoon season], Dutta et al.
[2019]). POC and PN were well correlated, which suggests that they shared the same origin (Fig.
S4, supplementary material). Despite evidence of DOC addition in the system in our study, the
low concentrations might result from rapid transport towards the coastal sea, as previously
discussed in detail for DIC.

By comparing the results of two POM mixing models (with and without the N/C ratio)

obtained separately from C, IB, and MR data, it is clear that the contribution of POM from
mangroves gradually decreased from C to IB to MR, and at the same time there was an overall
lack of riverine freshwater contribution to the bulk POM. In contrast, the contribution of the
seawater end-member (marine phytoplankton) increased gradually in the same fashion. This
result indicates that mangroves in this region do contribute to the water-column POM; however,
it is quickly transported from the narrow mangrove creeks to the river and subsequently to the
coastal sea. The mixing model using $\delta^{13}C$, $\delta^{15}N$, and N/C revealed that the mangrove
contribution is low even in the creeks (median value, 17%; range, 2–44%). The mixing model
using only $\delta^{13}C$ and $\delta^{15}N$ (Table S4) revealed that in the creeks, the input of mangrove-derived
POM was 33% (6–69%), which is comparable to values from other studies in the Sundarbans (0–
65.3%; Ray et al., 2015). These results further indicate that if we only consider the model with
$\delta^{13}C$ and $\delta^{15}N$, then previous studies in the Sundarbans might have overestimated the mangrove
contribution to POM.

A unique feature of the DOM in the present study is that most of the DOC values plotted

above the conservative mixing line, indicating its addition within the system. In contrast, the
optical indicators $a_{CDOM(254)}$, $a_{CDOM(375)}$, and $SUVA_{254}$ showed that almost all bulk CDOM plotted



below the conservative mixing line, indicating removal by the system. Both $a_{CDOM(254)}$ and
$a_{CDOM(375)}$ have been used in previous studies as metrics for the concentration of terrestrial
aromatic compounds (Weishaar et al., 2003; Zurbrugg et al., 2013) and as proxies for potential
refractory OM (Saadi et al., 2006; Hur et al., 2009). These optical signatures showed an overall
decrease with increases in salinity, i.e. from C towards MR. This result indicates an overall
decrease in the aromaticity of the DOM from the narrow creeks towards the coastal sea, whereas
Bergamaschi et al. (2012) reported that mangroves exude aromatic DOM. $S_{275-295}$ is used as an
optical indicator of potential terrestrial origin of CDOM (Fichot and Benner, 2012). Spencer et
al. (2012) suggested that increasing $S_{275-295}$ values are related to the decrease in molecular
weight and aromaticity of CDOM from fresh to marine waters. Almost all of the $S_{275-295}$ values
in the present study plotted above the conservative mixing line, indicating a lack of DOM of
freshwater origin. Moreover, the $S_{275-295}$ values in the present study were substantially higher
than the values reported by Ray et al. (2018b) for a mangrove environment in French Guiana
(South America), signifying less aromaticity of DOM than the typical mangrove-derived DOM.
**4.5 Seasonal and spatial up-scaling and comparison with global average $CO_2$ flux data**

We used the dataset of Akhand et al. (2016) for spatial and seasonal up-scaling and

updating of $CO_2$ flux data. They reported pCO$_2$(water) and $CO_2$ flux data for each month
annually and presented pre-monsoon (February–May), monsoon (June–September), and post-
monsoon (October–January) data from the upper, middle, and outer estuary of the Matla River.
Post-monsoon $CO_2$ flux was higher by a factor of about 9.6 and 13.3 in the middle and upper
estuary, respectively, than in the outer estuary. Similarly, the $CO_2$ flux was lower by a factor of
4.5 in the pre-monsoon season and 1.4 times higher in the monsoon season than in the post-
monsoon season. According to Akhand et al. (2016), the mean post-monsoon $CO_2$ flux in the



outer estuary was –44 µmol $CO_2$ m$^{-2}$ h$^{-1}$, whereas in the present study (also post-monsoon and
outer estuary) the mean $CO_2$ flux over all eight stations was –39.8 µmol $CO_2$ m$^{-2}$ h$^{-1}$. We used
the ratio of the flux value of Akhand et al. (2016) and the present study to up-scale and update
the $CO_2$ flux data for the whole estuary and for all seasons (throughout the year).
There are substantial differences between the methods used by Akhand et al. (2016) for
estimating and measuring $pCO_2$(water) and those in the present study, as well as between the
spatial and temporal resolution of the data. In addition, there might be differences between the
two studies due to interannual variation in the patterns of $pCO_2$(water) and other hydrological,
biogeochemical and meteorological conditions. Thus, the up-scaled $CO_2$ flux data should not be
used for their absolute values, which might have a higher uncertainty, but they can be useful for
comparison with globally averaged flux data.
On the basis of these criteria, we estimated the up-scaled $CO_2$ flux for the entire Matla
Estuary as 6.15 mmol m$^{-2}$ d$^{-1}$, which is much less than the world average for emissions from
mangrove-associated water (56.8 ± 8.9 mmol m$^{-2}$ d$^{-1}$; Rosentreter et al., 2018). We also
estimated the yearly $CO_2$ emission from the mangrove-associated waters of the Indian
Sundarbans using an area of 891 km$^2$, encompassing freshwater domains closed to blind rivers
and creeks in the Sundarban biosphere of India (Dubey et al., 2015) as $88 \times 10^6$ kg $CO_2$ y$^{-1}$.
**5. Conclusions**
Our results suggest that the water around mangroves can sometimes act as a sink or a
weak source for $CO_2$, contrary to most previous studies. Previous studies in the Sundarbans made
similar observations, but the precision and temporal resolution of the data were too coarse to
determine the source/sink characteristics. The present study successfully overcame the problems





and uncertainties with the earlier data by analysing the diel variability of pCO$_2$(water) at eight
sites covering the tidal maxima and minima and minimizing the possibility of over- or under-
estimation of CO$_2$ fluxes.

The elemental, stable isotopic, and optical signatures showed that the mangrove

ecosystem of the Sundarbans can contribute TAlk, DIC, DOC, and POC to the adjacent water (as
observed in other mangrove-associated waters around the globe); however, these carbon forms
are diluted by and transported to the pCO$_2$-lean coastal waters of the BoB. Moreover, stable
isotopic and optical signatures of this mangrove water showed negligible contributions from
riverine freshwater to the carbon input of this region. The minimal contribution of freshwater
input along with the predominance of pCO$_2$-lean northern BoB water increases the buffering
capacity, which, in turn, can explain the CO$_2$ sink properties of the waters surrounding the
Sundarban mangroves (or the reduced CO$_2$ efflux compared to global mangrove waters). Thus,
we argue that areas with such low emissions should be given due emphasis when up-scaling the
global mangrove carbon budget from regional observations.

**Data availability.** All data are available in this paper and supplementary material.

**Author contributions.** AA, AC, KW, SH, and TK designed the study. AA, AC, and SD did the
field work and sample collection. AA, AC, KW, SD, TT, and KC did the chemical and data
analysis. AA prepared the manuscript with input from AC and KW. SH and TK edited the
manuscript and provided all infrastructural facilities for this work. All authors read and approved
the manuscript.




**Competing interests.** The authors declare that they have no conflict of interest.


**Acknowledgements**

We are grateful for the funding of part of this work through the Environmental Research and Technology Development Fund (S-14) and the Japan Society for the Promotion of Science (KAKENHI 18H04156 and 19K20500). We also thank the National Remote Sensing Centre (Department of Space, Government of India) for partial funding of this work. We also thank the West Bengal Forest Department for granting us the necessary permissions. We are indebted to Ms. N. Umegaki for help with the chemical analyses.

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




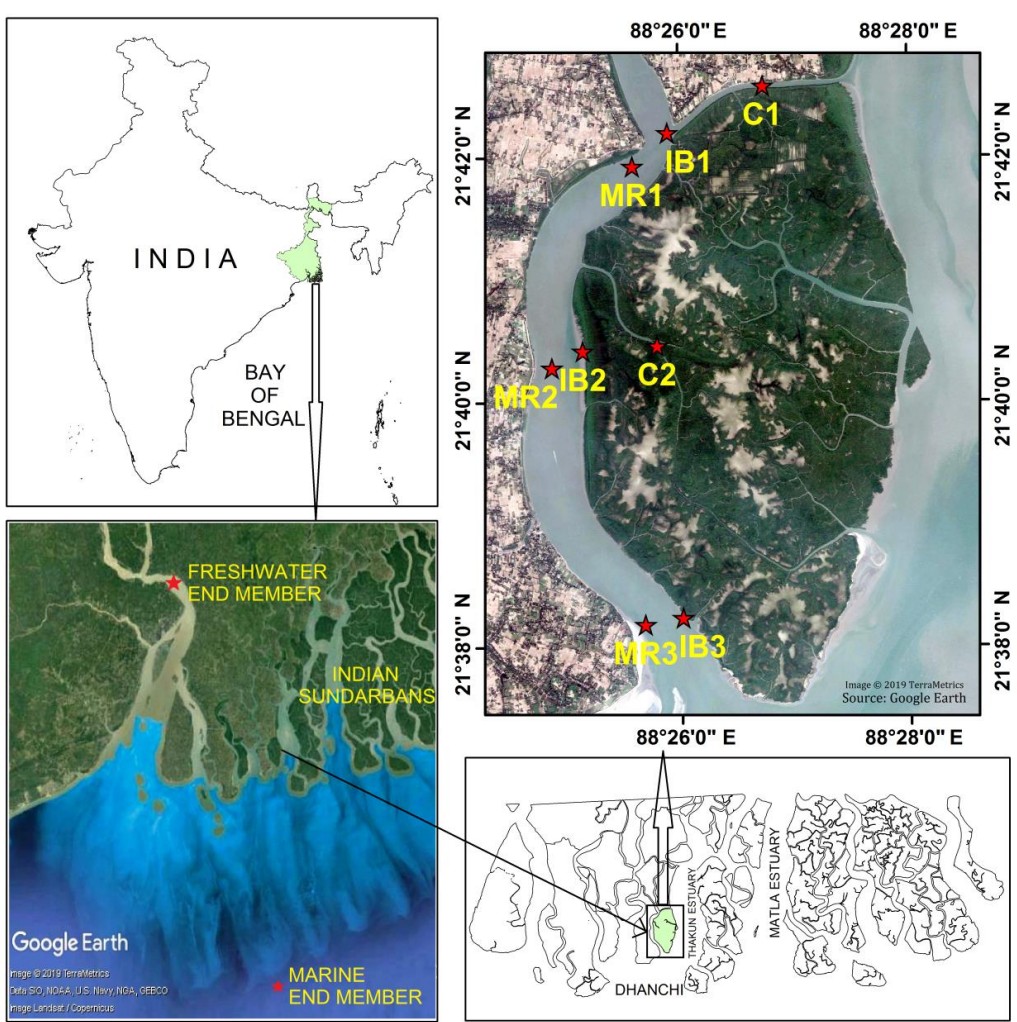


**Fig. 1.** Maps of the study area and of Dhanchi Island, located in the Indian part of the Sundarban

mangroves. The red stars denote the sampling locations within the two creeks (stations C1 and

C2), the three island boundary stations (IB1, IB2, and IB3) and the three mid-river stations

(MR1, MR2, and MR3), along with the stations for the freshwater end-member (FWEM) and

marine end-member (MEM) sample collections.

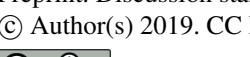



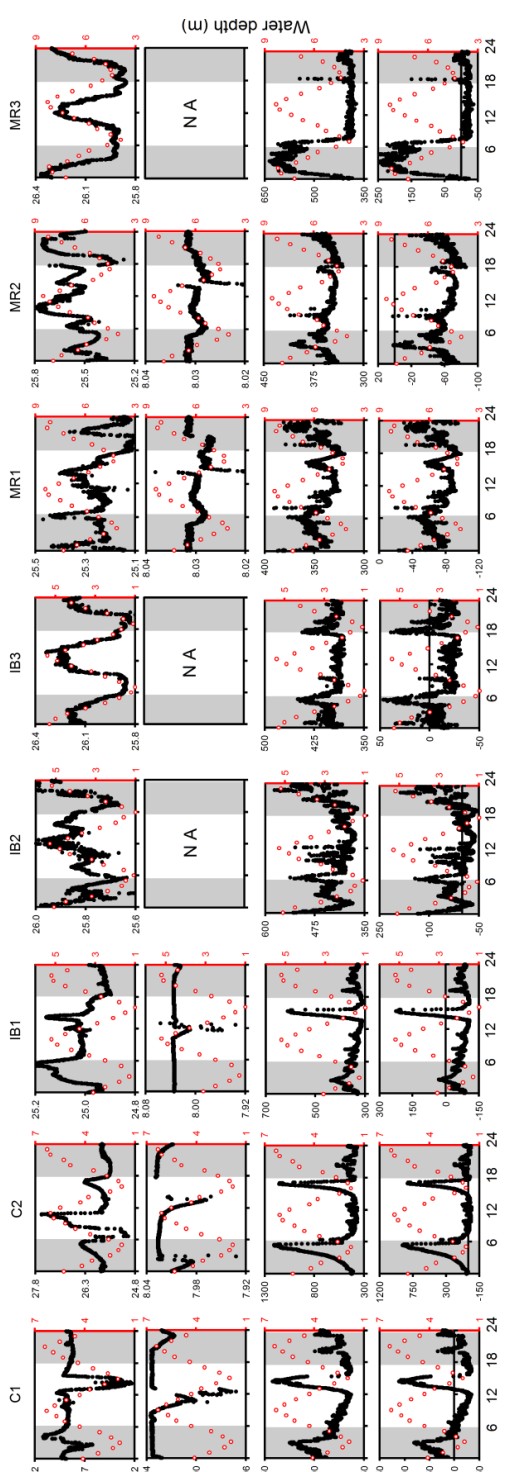

**Fig. 2** Time series plots of diel variation (presented in hour) of salinity, pH, pCO$_2$(water) (µatm), and CO$_2$ flux (µmol m$^{-2}$ h$^{-1}$), at the two creek stations (C1 and C2), the three island boundary stations (IB1, IB2, and IB3) and the three mid-river stations (MR1, MR2, and MR3), along with the variation in water depth (metres, shown on secondary y-axis). NA denotes that data could not be acquired because of unavoidable circumstances. The shaded regions show the night-time of the diel cycle.





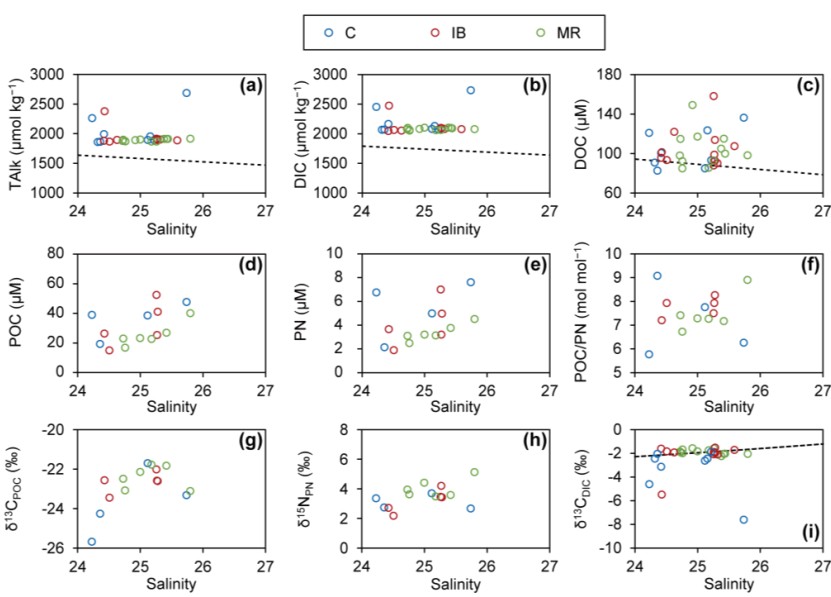

1047

**Fig. 3** Distribution of (a) TAlk, (b) DIC, (c) DOC, (d) POC, (e) PN, (f) POC:PN, (g) $\delta^{15}N_{PN}$, (h)

$\delta^{13}C_{POC}$, and (i) $\delta^{13}C$ of DIC, as a function of salinity, at creek (C), island boundary (IB), and

mid-river (MR) stations. The predicted two-end-member conservative mixing distributions of the

respective parameters are shown as dotted lines, assuming the following measured values for the

freshwater end-member (FWEM) and the marine end-member (MEM): FWEM TAlk = 2977 µM

kg$^{-1}$, MEM TAlk = 1646 µM kg$^{-1}$; FWEM DIC = 2950 µM kg$^{-1}$, MEM DIC = 1476 µM kg$^{-1}$;

FWEM DOC = 218.0 µM, MEM DOC = 79.2 µM; FWEM $\delta^{13}C_{DIC}$ = −6.6‰, MEM $\delta^{13}C$ of DIC

= −1.2‰.

1056

1057

1058



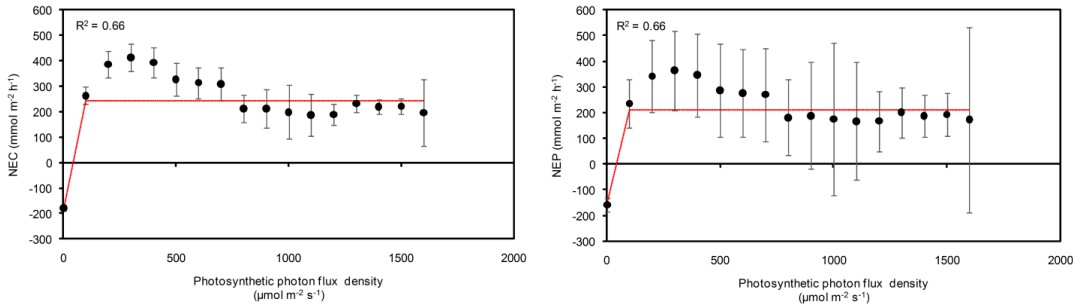

**Fig. 4** Bin average net ecosystem calcification (NEC) and net ecosystem production (NEP)

values at C2 plotted against photosynthetic photon flux density. The red lines show the

photosynthesis–irradiance curve fitted to measured NEP and NEC, and is statistically significant

($P < 0.0001$). The $R^2$ values are displayed on the respective plots. Error bars indicates standard

deviation.





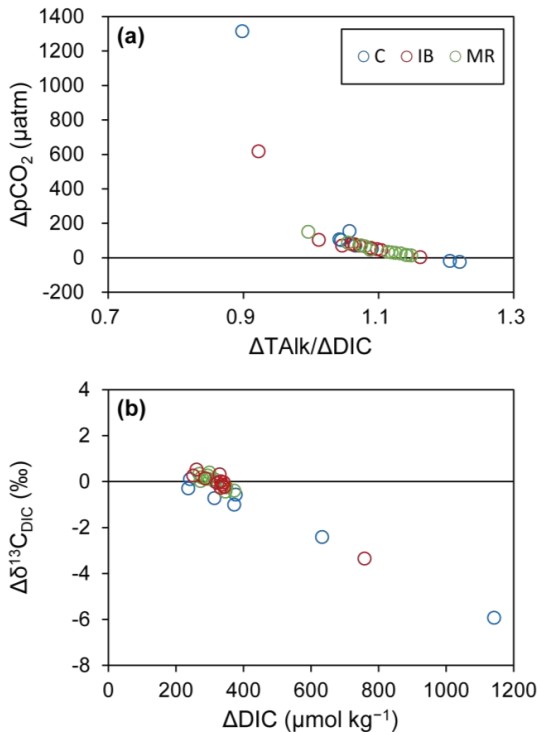

**Fig. 5. (a)** Relationship between the $\Delta$TAlk:$\Delta$DIC ratio and $\Delta$pCO$_2$ [the difference between the
in situ value of pCO$_2$(water) estimated using measured TAlk and DIC and the conservative
mixing value for the same]. **(b)** Deviation plot ($\Delta$DIC vs. $\Delta\delta^{13}C_{DIC}$).




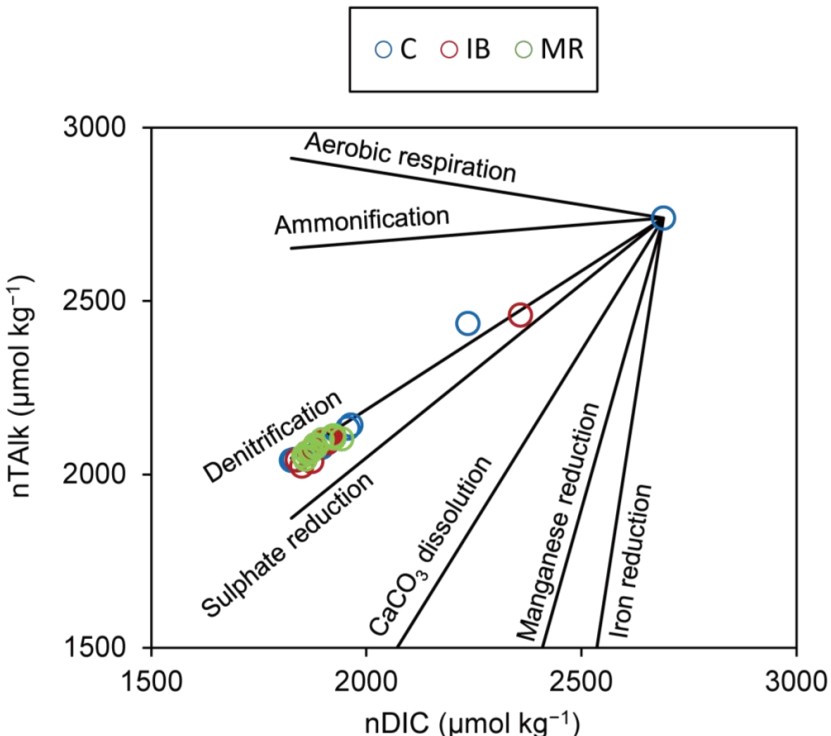


**Fig. 6.** Salinity-normalized TAlk (nTAlk) vs. salinity-normalized DIC (nDIC). The lines

correspond to the theoretical covariation of nTAlk and nDIC for various metabolic pathways,

and the coloured circles indicate data from the present study.














**Table 1** pCO$_2$(water) and air–water CO$_2$ flux observed in this study, and from the most recent
studies that measured pCO$_2$(water) and/or air–water CO$_2$ flux directly from the mangroves
surrounding waters.

| Place | pCO$_2$(water) [µatm] | CO$_2$ flux (mmol CO$_2$ m$^{-2}$ d$^{-1}$) | Remarks | Authors |
|---|---|---|---|---|
| North Brazil | 592 to 15,361 | 174 ± 129 | Spring to neap tidal cycle | Call et al. (2019) |
| Ouemo archipelago, New Caledonia | NA | 11 to 1620 | Dec, 2016 to Sept, 2017 (three week interval) only during high tide sessions | Jacotot et al. (2018) |
| Vietnam | 660 to 5000 | 74 to 876 | 24 h cycle in dry and wet season | David et al. (2018) |
| Queensland, Australia | 387 to 13031 | 58.7 to 277.6 | Wet season and dry season | Rosentreter et al. (2018) |
| Bali, Indonesia | NA | 18.1 ± 5.8 | 55 h time series measurement | Macklin et al. (2019) |
| Iriomote Island, Japan | 394 to 2667 | 9.1 ± 11.3 | July, 2017 | unpublished data, Akhand et al. |
| Sundarban, India | 311 to 1204 | –0.7 ± 1.7 to 1.6 ± 4.3 | Two weeks of post-monsoon (dry) season | Present study |
