# Peer review of "Low CO2 evasion rate from the mangrove surrounding waters of Sundarban"

_Biogeosciences, 2019_

## Referee Comment (RC1) · Anonymous Referee #1 · 5 Nov 2019

Review of Akhand et al submitted to BG

This paper reports some original new data in the water surrounding the Sundarban, the largest mangrove forest in the world. The major result obtained is the low pCO2 in these waters, which, contrarily to previous reports elsewhere, act as a CO2 sink to a moderate CO2 source. Although the pCO2 data presented here are apparently of the required quality, some other parameters (for instance alkalinity) have very unusual values, which makes publication hazardous. Most importantly, the paper fails at explaining why the mangrove waters in the Sundarban emit little CO2, and why these mangroves behave differently from the others elsewhere. The paper is very speculative and not based on adequate knowledge and appropriate reference to the literature, although the reference list is well updated. Some of the presented data are almost out of scope and

do not provide any relevant information to explain the observed low pCO2; some parameters such as NEP and NEC (Fig.4) are not described in the methods. Discussion and conclusion are very speculative and not supported by the data, nor by a correct analysis of the literature. For all these reasons, I recommend rejection.

Major comment #1. Data quality and data relevancy. In fig.2, the temporal evolution of parameters during 24h cycles show minimum pH (when available) and maximum pCO2 that are not in phase. Minimum pH always occurs about 2-3 hours before the maximum pCO2. The quality of these data is highly questionable. In addition, the authors report a total alkalinity (TA) value of 1.646 mmol kg-1 in the marine end-member (P14 L301), which is very strange. TA is very relatively constant in the surface ocean with values around 2.2 mmol kg-1 (same values as those reported by Goyer et al.). Even if the salinity of this marine end-member is only 26, if the freshwater TA is 2.977 (P17 L 361), it is hard to believe that TA at salinity 26 can be so low... The authors present data of Net ecosystem Production and Net Ecosystem Calcification, but the methods are not described in the main text. In addition, they present the result of these parameters ONLY FOR ONE STATION, because "measured NEP and NEC yielded statistically significant relationships only at C2 among the eight stations" (P18L382). If I well understood, the method gave exploitable results only at one station and incoherent or not significant results at the other 7 stations. If this is the case, I suggest the authors question the reliability of their method and simply do not publish the results of the unique case were it apparently worked before checking what was wrong with the other 7 stations.

Major comment #2. The authors attribute low pCO2 to the buffer capacity of the carbonate system in these waters. However, the buffer capacity alone will never make seawater change from CO2 source to CO2 sink. Buffer capacity (high revelle factor) will make the pCO2 lower for a same CO2 input, but it will never make the water a sink. Biological uptake is necessary (in the case here of a high alkalinity in the freshwater end-member, thermodynamical mixing will not generate pCO2 values below the

atmospheric equilibrium). The authors also attribute low pCO2 to strong "dilution" of mangrove soil porewater with estuarine surface waters. However, they do not provide quantitative evidence that dilution could be more important in the Sundarban than in others mangrove waters elsewhere in the world. All the discussion is extremely speculative and finally it does not explain why pCO2 is low at the study site. A detailed analysis of pCO2 variations as a function of salinity may have made the paper less speculative.

Detailed comments P11: it is not clear what is the interest of these CDOM and SUVA data for the main message of the paper (same for NEP and NEC) P13 L 268: why choosing the k600 of Ho et al. 2011 and not other parameterizations from other authors? P13L281 NEP and NEC are not easy to measured and it is strange to describe these methods in the supplementary material P14L286 how is Xmix calculated? Provide a formula L294 what is a "near-zero salinity regime"? L292-299 this looks like discussion and not material and method L300-312 this looks like results or discussion but not material and method P18 section 3.4 these parameters are not defined and explained in the text. The fact that only one value could be obtained among the 8 experiment make the quality of these data questionable. Why are NEP and NEC negative? This is not understandable. Section 3.5. the problem with these parameters is that they do not provide relevant information that could help the interpretation of the low pCO2 values. ARE THESE WATERS RICH IN PHYTOPLANKTON? Uptake of CO2 by phytoplankton could be the reason of low pCO2 L398-404: only truisms here, you say basically nothing Section 4.2: the term "diel" is confusing here, as most of the pCO2 variations are driven by tidal movements, but "diel" generally refer to differences between night and day. The term "CO2-lean seawater" you use all through the MS is awkward

L437 this is speculation, not supported by any data L439 "as the consumption of bicarbonate by phytoplankton during photosynthesis activity tends to convert pCO2(water) to bicarbonate ions (ie a decrease in pCO2)" please write statements in accordance

with classical handbooks on ocean carbonate chemistry. L447-448 "the photosynthetic potential and composition of phytoplankton would not vary widely between these stations because they are close to each other" this is just speculation, no data on phytoplankton are available The all section L446-453 is pure speculation L462 "the unit increase in pCO2 with respect to the unit input of DIC was much higher at the creek stations than at the IB station" not understandable L465 "greater water turnover enhances the solute fraction" any quantitative evidence for that? L467-468 high speculatives statements L474 last sentence "pore water is generated exclusively within the mangrove environment" awkward formulation L474 groundwater seepage would increase pCO2 not decrease End of page 22 until centre of page 23 is pure speculation, the revelle Factor will not explain explain low pCO2 in this cases

L505: buffering capacity cannot generate a CO2 sink alone... L513 high d13C-DIC values are not necessarily due to carbonate dissolution, it can be fractionation by phytoplankton L508-520 the potential impact of phytoplankton and gas exchange on d13C-DIC is missing Section 4.4 is highly speculative and not connected to the question of low pCO2

Fig3 : use squares triangles and circles as symbols; an explanation for the positive TA anomalies is missing in the text

---

## Referee Comment (RC2) · Anonymous Referee #2 · 23 Dec 2019

The present study investigates water pCO2 at 8 different stations of mangrove surrounding waters (creek, island boundary, mid-river) at Dhanchi Island in the Sundarbans, India. The authors present an interesting high resolution data set (8x 24h time series, diurnal, tidal) at 1 min interval of pCO2 and find mangrove surrounding waters to be a weak source or sink of atmospheric CO2. The authors aim to reveal and identify why the here studied mangrove waters act as a net sink compared to previous studies, that are commonly found to be a source of CO2. They conclude that the reduced riverine input and increased buffering capacity from oceanic water is responsible for the low pCO2 in the mangrove waters.

Although the data set is impressive and worthwhile publication, I am not convinced that the authors have sufficiently identified and discussed the low pCO2 at the different

study locations based on their data. The discussion is very speculative (see reviewer #1, I mostly agree with reviewer #1: regarding the low TAlk of the marine end-member. This value is questionable. I also agree, that the Revelle cannot be used to explain the CO2 sink. I further agree with reviewer #1 that some of the data (optical, NEP, NEC) seem out of context and do not provide relevant information to explain the low pCO2. NEP and NEC calculations need to be included in the methods section. High salinity combined with high abundance of phytoplankton or benthic micro-algae could be an explanation for the low pCO2.)

The authors mention that there is no (or almost none) riverine connection. Yet, they use a freshwater end-member upstream to estimate the conservative mixing lines, which does not makes sense if there is no riverine connection. Similarly, the marine end-member seems questionable with a salinity of 26 , which is very close to the mangrove waters (salinity 25-26).

Secondly, the station C1 and C2 are substantially different and should not be treated as one group. To me, station C2 seems like the only "real" mangrove site. As in several other previously studied mangrove surrounding water locations cited in this manuscript, a single creek ending in a mangrove forest is the ideal location to study tidal and temporal variability and fluxes of inorganic carbon and dissolved gases (tidal pumping). C2 has no connection other than to the estuary. In contrast, C1 is not a "creek" but more a branch or tributary of the main estuary channel that connects the left and right (Thakun) estuary channels, therefore is influenced by biogeochemical processes of both channels. I disagree that 20 meter width is indicative of a "very narrow creek". I am not surprised to see the very low pCO2 in the main estuary (not river) channel and closeby island boundary. These study sites (MR, IB, C1) seem more indicative of a marine environment with low change in salinity (salinity 24-27). What is the effect from macro-tides compared to meso or micro-tides ? Fig.2 MR3 shows a typical diurnal trend of CO2 rather than a tidal trend. The authors identified correctly that the term "mangrove surrounding waters" can be ambiguous. Station

MR1-3 might be better to compare to (previous) estuary $CO_2$ emissions than mangrove $CO_2$ emissions?

The latest global mangrove forest distribution is Bunting et al. (2018) Bunting P, Rosenqvist A, Lucas RM, Rebelo LM, Hilarides L, Thomas N, Hardy A, Itoh T, Shimada M, Finlayson CM. 2018. The global mangrove watch - a new 2010 global baseline of mangrove extent. Remote Sensing 10 (10) DOI: 10.3390/rs10101669

L97-98 what is the difference between "mangrove surrounding waters" and "mangrove waters" in this context here? I would suggest to define what you mean with "mangrove surrounding waters" at the beginning of the manuscript and then use this term consistently throughout the manuscript.

I would suggest to change the title. The term "evasion rate" implies an efflux of $CO_2$ from water to the atmosphere while the authors aim to highlight the influx. Alternatively, a title similar to this : "Low $pCO_2$ in mangrove surrounding water in the Sundarbans".

The gas transfer velocity is the highest uncertainty in the gas flux computation, therefore k parameterisations should be chosen carefully. It is advisable to compare fluxes based on several different k parameterisations (not just one) in dynamic tidal ecosystems such as mangrove estuaries. It would be interesting to see how much this would change the average influx/efflux.

L498 "$pCO_2$ concentration" is wrong. It is $pCO_2$ or $CO_2$ concentration (e.g. $\mu$M).

L521-529: This is unclear. Do the authors suggest that the source of DIC is a mix of all the possible sources listed in this paragraph?

L436-439, L601 The authors suggest "rapid transport to the coastal ocean". Do they mean rapid flushing of pore water? Or tidal pumping? Why rapid dilution? This is unclear. It might be helpful to calculate the freshwater flushing times for the estuary to support this hypothesis. Although with no or very little riverine input I assume very low flushing.

Yes, the term "pCO2-lean seawater" is awkward.
* * *

---

## Author Comment (AC1) · 10 Jan 2020

Response to reviewer 1's comments:

Reviewer 1's comments: This paper reports some original new data in the water surrounding the Sundarban, the largest mangrove forest in the world. The major result obtained is the low pCO2 in these waters, which, contrarily to previous reports elsewhere, act as a CO2 sink to moderate CO2 source. Although the pCO2 data presented here are apparently of the required quality, some other parameters (for instance alkalinity) have very unusual values, which makes publication hazardous. Most importantly, the paper fails at explaining why the mangrove waters in the Sundarban emit little CO2, and why these mangroves behave differently from the others elsewhere. The paper is
very speculative and not based on adequate knowledge and appropriate reference to the literature, although the reference list is well updated. Some of the presented data are almost out of scope and do not provide any relevant information to explain the observed low pCO2; some parameters such as NEP and NEC (Fig.4) are not described in the methods. Discussion and conclusion are very speculative and not supported by the data, nor by a correct analysis of the literature. For all these reasons, I recommend rejection.

Major comment #1: Data quality and data relevancy. In fig.2, the temporal evolution of parameters during 24h cycles show minimum pH (when available) and maximum pCO2 that are not in phase. Minimum pH always occurs about 2-3 hours before the maximum pCO2. The quality of these data is highly questionable. In addition, the authors report a total alkalinity (TA) value of 1.646 mmol kg-1 in the marine end-member (P14 L301), which is very strange. TA is very relatively constant in the surface ocean with values around 2.2 mmol kg-1 (same values as those reported by Goyer et al.). Even if the salinity of this marine end-member is only 26, if the freshwater TA is 2.977 (P17 L 361), it is hard to believe that TA at salinity 26 can be so low: : : The authors present data of Net ecosystem Production and Net Ecosystem Calcification, but the methods are not described in the main text. In addition, they present the result of these parameters ONLY FOR ONE STATION, because "measured NEP and NEC yielded statistically significant relationships only at C2 among the eight stations" (P18L382). If I well understood, the method gave exploitable results only at one station and incoherent or not significant results at the other 7 stations. If this is the case, I suggest the authors question the reliability of their method and simply do not publish the results of the unique case were it apparently worked before checking what was wrong with the other 7 stations.

Authors' Response: Thank you very much Reviewer 1 for the constructive comments on the improvement of our manuscript.

(1) Quality of the data:

We believe that the quality of the data was thoroughly maintained for all the parameters. However, thanks to Reviewer 1's comments, we found an unintentional error in our present version of the manuscript. Almost all the sensors used in this study were set according to Japan Standard Time. We prepared the other graphs by correctly adjusting the data to Indian Standard Time, but unfortunately, we wrongly adjusted the time stamp for our pH sensor by mistake. As Indian Standard Time and Japan Standard Time has a time difference of three and a half hours, this disparity between pH and pCO2(water) data occurred. We'll correct the time stamp for our pH sensor in the revised manuscript. Nevertheless, any interpretation has not been affected for this unintentional error.

(2) Low TA in MEM during the post-monsoon season:

We agree that the TA value in the marine end member (MEM) is unusually low. That is why we have already explained the background of MEM in details in the methodology section (from Line 300 to 312) using published and our unpublished data. Briefly, the TA data of the present study are also in good agreement with a few available published work in this transition zone conducted during the post-monsoon season of 2010-2011 and 2011-2012 (January and February) (Akhand et al. 2012; 2013), showing low values during post monsoon season (the season of the present study). From the unpublished data, the TA of MEM was found to be higher during pre-monsoon (1932 $\mu$mol kg-1) and monsoon (1879 $\mu$mol kg-1), than post-monsoon (1646 $\mu$mol kg-1). Indeed, the sampling and cruise of Goyet et al. (1999) was conducted between 29 August 1995 and 16 October 1995, which was of different season and far offshore (approx. 10o N latitude). Furthermore, the present study used the standard preservation method for TA samples and used the gran plot method with an automatic titrator (batch sample analyser). Moreover, a certified reference material was procured and used to maintain the company specific accuracy of the batch sample analyser. Therefore, we believe that the present work is reporting sufficient precision to be published. To further examine the interesting phenomena of low TA in MEM during the post-monsoon season, we

also examined the modelled data archive (https://esgf-node.llnl.gov/projects/esgf-llnl/) (Dunne et al. 2012). The spatial resolution of the data is coarse, i.e. 1o; hence, not exactly comparable with our field observation data because these data covers whole area of the coastline to the offshore transition zone, but we believe that it can be used as a supporting data to understand the carbonate chemistry of MEM. We used the spatial extent of 20o to 21oN and 87oto 91oE to extract the data which covers a substantial area of the northern Bay of Bengal and for the years 1995-1996 to 2004-2005 (10 consecutive years). The extracted data showed the mean TA value being 2.11 mmol kg-1(pre-monsoon, from February to May), 2.04 mmol kg-1(monsoon, from June to September) and 1.98 mmol kg-1(post-monsoon, from October to January), indicating the same pattern as our field observation data, i.e., TA never reaches 2.2 mmol kg-1 in the transition zone, and both TA and DIC decrease during post-monsoon season. The decrease in both TA and DIC during post-monsoon seasons might be because of phytoplanktonic calcifies (foraminifera and coccolithophores) blooming during post-monsoon season (Biswas et al. 2004) in the northern Bay of Bengal (Stoll et al. 2007; Mergulhao et al. 2013).

Overall, the phenomenon of low TA in MEM during the post-monsoon season is indeed a very interesting topic of research. However, addressing further this phenomenon is out of scope for our present work.

(3) NEP and NEC:

We agree with the comments of both Reviewer 1 and 2. We'll eliminate all the part of NEP and NEC in the revised manuscript, according to your suggestion.

Authors' changes in the manuscript: We'll correct the time stamp of the pH data in Fig. 2 according to Indian Standard Time and will eliminate NEP and NEC part from the whole manuscript. We'll add sentence(s) and citation(s) to describe the probable reason behind such low magnitudes of TAlk and DIC during post-monsoon in the marine end member in 'Data Analysis' (section 2.6) under 'Materials and methods' while

describing the carbonate chemistry of the MEM.

Comment of the Reviewer 1:

Major comment #2: The authors attribute low pCO2 to the buffer capacity of the carbonate system in these waters. However, the buffer capacity alone will never make sea water change from CO2 source to CO2 sink. Buffer capacity (high revelle factor) will make the pCO2 lower for a same CO2 input, but it will never make the water a sink. Biological uptake is necessary (in the case here of a high alkalinity in the freshwater end-member, thermodynamical mixing will not generate pCO2 values below the atmospheric equilibrium). The authors also attribute low pCO2 to strong "dilution" of mangrove soil porewater with estuarine surface waters. However, they do not provide quantitative evidence that dilution could be more important in the Sundarban than in others mangrove waters elsewhere in the world. All the discussion is extremely speculative and finally it does not explain why pCO2 is low at the study site. A detailed analysis of pCO2 variations as a function of salinity may have made the paper less speculative.

Authors' Response: The focus of the present work is not 'CO2 sink character'. It is well established fact, as a whole, mangrove surrounding water of Sundarban (Indian part) is net source of CO2 covering all seasons throughout the year and considering upper to lower estuary. The fact is evidenced from a number of previous studies and in parity with the upscaled data of the present study (section 4.5). The tittle of the present manuscript also represents CO2 'evasion', not 'sink'. Hence, we explained in the present version of the manuscript that the CO2 efflux / evasion rate of Sundarban is much lesser than the recently estimated world average due to high buffer capacity (low revelle factor).We also agree with that biological uptake can also an important mechanism explaining for low pCO2.

However, we did not state about biological uptake in the present manuscript because biological uptake seems to be minor. We showed using Fig. 5a that most of the ∆pCO2

value was positive, which indicates the study area exhibited heterotrophy. We'll add sentence(s) for better clarification of this issue. Furthermore, we collected time series phytoplankton standing stock (chlorophyll-a fluorescence) data using a fluorometer but did not get any significant correlation with pCO2 (water). However, we agree with the reviewer's concern and will present the time series data of chlorophyll-a in the supplementary material (to be included in figure S1). We'll also add discussion about less correlation with pCO2 (water) and chlorophyll-a to show the less significant effect of phytoplankton productivity in pCO2(water) and CO2 flux, especially in the stations showing CO2 sink.

Nevertheless, we believe the main reason for the low pCO2 character is mainly twofold: 1. Special character of MEM i.e. Bay of Bengal and subsequent predominance of low pCO2 marine water. Bay of Bengal is specially having low pCO2 and considered as a sink for CO2 (Kumar et al. 1996; Goyet et al. 1999; Akhand et al. 2013). In contrast, western part of Indian Ocean, i.e. Arabian sea is considered as a strong source of CO2 (Kortzinger and Duinker 1997; Sarma et al. 1998; Sarma 2003). 2. Getting lesser amount of riverine freshwater, which are well discussed.

In turn, Reviewer 1's notion that we are stating that "quantitative evidence that dilution could be more important in the Sundarban than in others mangrove waters elsewhere in the world" is a misunderstanding as we have not stated such an assertion. We believe, dilution is equally important in all mangrove surrounding water, that is why tidal variability i.e. high tide / low tide shifting is so important explaining CO2 dynamics in mangrove surrounding water (Ovalle et al., 1990; Maher et al., 2013; Akhand et al. 2016; Yang et al. 2017). Hence, the tidally driven estuaries of central Sundarban (Indian part) are affected by the low pCO2 water of Bay of Bengal. This effect of Bay of Bengal, is further accentuated for rapid transport of material due to estuarine geometry, as discussed in L 530 to L 535 of the present version of the manuscript. However, according to Reviewer 1 and Reviewer 2's concern and to avoid ambiguity, we'll delete the word 'rapid' from L 436 and L 437 in the revised manuscript.

Authors' changes in the manuscript: We'll add / edit sentence(s) for better understanding of the Revelle factor issue. We'll add a separate paragraph to clarify about the 'Biological factor' in relation with low pCO2. We'll add sentence(s) on the heterotrophic nature of the study area showing Fig. 5a, regarding $\triangle$pCO2 value. We'll add chlorophyll-a time series data in the Fig. S1 and will add sentence(s) on the relation between pCO2 and Chlorophyll-a in the 'Discussion' section along with necessary sentence(s) in the 'Materials and methods' and 'Results' section. We'll add sentence(s) and references(s) on the special low pCO2 character of the Bay of Bengal in the 'Introduction' section. We'll delete the word 'rapid' from L 436 and L 437.

Comment of the Reviewer 1:

Detailed comments: P11: it is not clear what is the interest of these CDOM and SUVA data for the main message of the paper (same for NEP and NEC)

Authors' Response: According to the suggestions of Reviewer 1 and Reviewer 2, we'll eliminate the NEP, NEC and optical signature, i.e. CDOM part from the whole manuscript.

Authors' changes in the manuscript: We'll eliminate the NEP, NEC and optical signature, i.e. CDOM part from the whole manuscript.

Detailed comments: L 268: why choosing the k600 of Ho et al. 2011 and not other parameterizations from other authors?

Authors' Response:

The justification is there in L 267 to L 271. However, according to the concern of Reviewer 1 and suggestion of Reviewer 2, we'll add other parameterisations for flux calculation.

Authors' changes in the manuscript: We'll add other parameterisations for CO2 flux calculation.

Detailed comments: NEP and NEC are not easy to measured and it is strange to describe these methods in the supplementary material

Authors' Response: We'll eliminate NEP and NEC part from the whole manuscript.

Authors' changes in the manuscript: We'll eliminate NEP and NEC part from the whole manuscript.

Detailed comments: how is Xmix calculated? Provide a formula

Authors' Response: For calculation of Xmix, linear and non-linear two end member mixing model have been implemented, which are in common use and the formula proposed by Mook and Tan (1991) have already been cited in the present version of the manuscript, we believe that no more details are needed.

Authors' changes in the manuscript: No change.

Detailed comments: what is a "near-zero salinity regime"?

Authors' Response: We'll replace the phrase "near-zero salinity regime" by "near-zero salinity region" in the revised manuscript.

Authors' changes in the manuscript: We'll replace the phrase "near-zero salinity regime" by "near-zero salinity region" in the revised manuscript.

Detailed comments: L292-299 this looks like discussion and not material and method L300-312 this looks like results or discussion but not material and method

Authors' Response: L 292 to L 299 of the present version of the manuscript is the justifications for choosing the freshwater end member and marine end member sites. Further, L300-312 is justifications to explain the characteristics of carbonate chemistry parameters of the marine end member and the Bay of Bengal, as the Reviewer 1 questioned. We believe that it is better remained this part in 'Materials and methods' section rather than 'Results' or 'Discussion'.

Authors' changes in the manuscript: No change.

Detailed comments: P18 section 3.4 these parameters are not defined and explained in the text. The fact that only one value could be obtained among the experiment make the quality of these data questionable. Why are NEP and NEC negative? This is not understandable.

Authors' Response: We'll eliminate NEP and NEC part from the whole manuscript.

Authors' changes in the manuscript: We'll eliminate NEP and NEC part from the whole manuscript.

Detailed comments: Section 3.5. The problem with these parameters is that they do not provide relevant information that could help the interpretation of the low pCO2 values. ARE THESE WATERS RICH IN PHYTOPLANKTON? Uptake of CO2 by phytoplankton could be the reason of low pCO2

Authors' Response: Despite of not being directly related, why OM related parameters are also important in relation to pCO2(water) and air-water CO2 flux in mangrove environment, have already explained thoroughly in L 575 to L 583.We'll add a separate paragraph and discuss about 'biological uptake' and phytoplankton standing stock (chlorophyll-a) in relation to pCO2(water) and could show the effect of 'biological uptake' and phytoplankton productivity on low pCO2 and CO2 uptake in this case.

Authors' changes in the manuscript: We'll add a separate paragraph and discuss about 'biological uptake'. We'll add chlorophyll-a time series data in the Fig. S1 and will add sentence(s) on the less correlation between pCO2 and Chlorophyll-a in the 'Discussion' section along with necessary sentence(s) in the 'Materials and methods' and 'Results' section.

Detailed comments: L398-404: only truisms here, you say basically nothing

Authors' Response: This part will be eliminated, as we'll eliminate the optical signature, i.e. CDOM part.

Authors' changes in the manuscript: We'll eliminate optical signature, i.e. CDOM part from the whole manuscript.

Detailed comments: Section 4.2: the term "diel" is confusing here, as most of the $pCO_2$ variations are driven by tidal movements, but "diel" generally refer to differences between night and day. The term "$CO_2$-lean seawater" you use all through the MS is awkward.

Authors' Response: As per our knowledge, "diel variability" refers to variability within "24 hours cycles which includes day and night shifts". We think, though the term 'diel variability' generally used to "refer the differences between day and night", but it can be also used for other differences within 24 hrs cycle (for example, due to high tide / low tide). However, for better clarification, we'll use 'diel and tidal variability' according to the purpose of use. "$pCO_2$ lean" will be replaced by "low $pCO_2$" in this section and throughout the manuscript.

Authors' changes in the manuscript: We'll use 'diel and tidal variability' according to the purpose of use in the revised manuscript. We'll replace the term "$pCO_2$ lean" by "low $pCO_2$" throughout the manuscript.

Detailed comments: L437 this is speculation, not supported by any data

Authors' Response: We believe that this is not speculation because it has been supported by DIC and $\delta 13CDIC$ data. Please see figures 3b, 3i and 5a. However, we'll delete the word 'rapid' from this sentence.

Authors' changes in the manuscript: We'll cite figures 3b, 3i and 5a in L437 and will delete the word 'rapid' from this sentence.

Detailed comments: L439 "as the consumption of bicarbonate by phytoplankton during photosynthesis activity tends to convert $pCO_2$(water) to bicarbonate ions (ie a decrease in $pCO_2$)" please write statements in accordance with classical handbooks on ocean carbonate chemistry.

Authors' Response: We agree with the reviewer, that this statement was a bit ambiguous and meaning was not clear. We'll revise the sentence in the revised manuscript.

Authors' changes in the manuscript: We'll revise the sentence in the revised manuscript.

Detailed comments: L447-448 "the photosynthetic potential and composition of phytoplankton would not vary widely between these stations because they are close to each other" this is just speculation, no data on phytoplankton are available

Authors' Response: We'll provide chlorophyll-a data as well as the relationship with pCO2 in the revised manuscript.

Authors' changes in the manuscript: We'll add chlorophyll-a time series data in the Fig. S1 and will add sentence(s) on the less correlation between pCO2 and Chlorophyll-a in the 'Discussion' section along with necessary sentence(s) in the 'Materials and methods' and 'Results' section.

Detailed comments: The all section L446-453 is pure speculation

Authors' Response: We'll delete this part from the revised manuscript as it is not directly related to the scope of the study.

Authors' changes in the manuscript: We'll delete this part from the revised manuscript.

Detailed comments: L462 "the unit increase in pCO2 with respect to the unit input of DIC was much higher at the creek stations than at the IB station" not understandable

Authors' Response: We wanted to convey the exact meaning of higher Revelle factor. Revelle factor actually tells us the unit increase in pCO2 when unit DIC is introduced to the aquatic system. However, we'll simplify the sentence and will re-write it in the revised manuscript.

Authors' changes in the manuscript: The sentence mentioned by the reviewer will be recast in the revised manuscript.

Detailed comments: L465 "greater water turnover enhances the solute fraction" any quantitative evidence for that?

Authors' Response: No, we do not have any quantitative evidence. We'll change the sentence and citation to explain the role of pore water on $pCO_2$ dynamics in mangrove surrounding water, as there is no quantitative evidence.

Authors' changes in the manuscript: We'll change the sentence and citation.

Detailed comments: L467-468 high speculatives statement

Authors' Response: The speculative part will be deleted.

Authors' changes in the manuscript: We'll delete the speculative part of the sentence.

Detailed comments: L474 last sentence "pore water is generated exclusively within the mangrove environment" awkward formulation

Authors' Response: We'll revise the sentence in the revised manuscript.

Authors' changes in the manuscript: We'll revise the sentence in the revised manuscript.

Detailed comments: L474 groundwater seepage would increase $pCO_2$ not decrease

Authors' Response: We have never asserted it.

Authors' changes in the manuscript: No change.

Detailed comments: End of page 22 until centre of page 23 is pure speculation, the revelle Factor will not explain explain low $pCO_2$ in this cases

L505: buffering capacity cannot generate a $CO_2$ sink alone.

Authors' Response: We agree with Reviewer 1 at this point that 'biological uptake' should be discussed for better clarification. We'll add this part and we'll add phytoplankton standing stock data, which can provide further explanation about low $pCO_2$.

However, we disagree with "the revelle Factor will not explain low pCO2" as we replied above both buffering capacity (Revelle Factor here) and biological uptake are important mechanisms explaining low pCO2. Authors' changes in the manuscript: We'll add sentence(s) to better explain revelle factor. We'll add a separate paragraph about the role of 'biological uptake' in low pCO2. We'll add chlorophyll-a time series data in the Fig. S1 and will add sentence(s) on the less correlation between pCO2 and Chlorophyll-a in the 'Discussion' section along with necessary sentence(s) in the 'Materials and methods' and 'Results' section.

Detailed comments: L513 high d13C-DIC values are not necessarily due to carbonate dissolution, it can be fractionation by phytoplankton

Authors' Response: In this line, we explained the deviation plot. Here, the determining factor is not only 'high d13C-DICvalues', but the combination of positive ΔDIC with positive Δd13C-DIC with respect to their conservative mixing concentration. We'll add sentence on the effect of isotopic fractionation by phytoplankton in appropriate place. However, in L513, 'fractionation by phytoplankton' is not the dominant regulating factor.

Authors' changes in the manuscript: We'll add sentence on the effect of isotopic fractionation by phytoplankton to appropriate place.

Detailed comments: L508-520 the potential impact of phytoplankton and gas exchange on d13CDIC is missing

Authors' Response: The potential impact of phytoplankton is mentioned in L 518. We'll add a sentence about the negligible effect of gas exchange, as the CO2 exchange from the water was nearly in equilibrium.

Authors' changes in the manuscript: We'll add a sentence about the negligible effect of gas exchange, as the CO2 exchange from the water was nearly equilibrium.

Detailed comments: Section 4.4 is highly speculative and not connected to the question of low pCO2

Authors' Response: The optical signature part, i.e. CDOM will be eliminated according to the suggestion of Reviewer 1 and Reviewer 2 . Authors' changes in the manuscript: We'll eliminate optical signature, i.e. CDOM part from the whole manuscript.

Detailed comments: Fig3: use squares triangles and circles as symbols; an explanation for the positive TA anomalies is missing in the text

Authors' Response: We'll edit Fig.3 according to the suggestion of Reviewer 1 and add sentence(s) on positive TA anomaly.

Authors' changes in the manuscript: We'll edit Fig.3 according to the suggestion of Reviewer 1 and add sentence (s) on positive TA.

References

Akhand, A., Chanda, A., Dutta, S. and Hazra, S., 2012. Air–water carbon dioxide exchange dynamics along the outer estuarine transition zone of Sundarban, northern Bay of Bengal, India. Indian Journal of Geo-Marine Science,41(2),pp.111-116.

Akhand, A., Chanda, A., Dutta, S., Manna, S., Hazra, S., Mitra, D., Rao, K.H. and Dadhwal, V.K., 2013. Characterizing air–sea CO2 exchange dynamics during winter in the coastal water off the Hugli-Matla estuarine system in the northern Bay of Bengal, India. Journal of oceanography, 69(6), pp.687-697. Akhand, A., Chanda, A., Manna, S., Das, S., Hazra, S., Roy, R., Choudhury, S.B., Rao, K.H., Dadhwal, V.K., Chakraborty, K. and Mostofa, K.M.G., 2016. A comparison of CO2 dynamics and air‐water fluxes in a river‐dominated estuary and a mangrove‐dominated marine estuary. Geophysical Research Letters, 43(22), pp.11-726.

Biswas, H., Mukhopadhyay, S.K., De, T.K., Sen, S. and Jana, T.K., 2004. Biogenic controls on the air—water carbon dioxide exchange in the Sundarban mangrove environment, northeast coast of Bay of Bengal, India. Limnology and Oceanography, 49(1), pp.95-101.

Dunne, J.P., John, J.G., Adcroft, A.J., Griffies, S.M., Hallberg, R.W., Shevliakova, E.,

Stouffer, R.J., Cooke, W., Dunne, K.A., Harrison, M.J. and Krasting, J.P., 2012. GFDL's ESM2 global coupled climate–carbon earth system models. Part I: Physical formulation and baseline simulation characteristics. Journal of Climate, 25(19), pp.6646-6665.

Goyet, C., Coatanoan, C., Eischeid, G., Amaoka, T., Okuda, K., Healy, R. and Tsunogai, S., 1999. Spatial variation of total CO2 and total alkalinity in the northern Indian Ocean: A novel approach for the quantification of anthropogenic CO2 in seawater. Journal of Marine Research, 57(1), pp.135-163. Körtzinger, A. and Duinker, J.C., 1997. Strong CO2 emissions from the Arabian Sea during south‐west monsoon. Geophysical Research Letters, 24(14), pp.1763-1766.

Kumar, M.D., Naqvi, S.W.A., George, M.D. and Jayakumar, D.A., 1996. A sink for atmospheric carbon dioxide in the northeast Indian Ocean. Journal of Geophysical Research: Oceans, 101(C8), pp.18121-18125.

Maher, D.T., Santos, I.R., Golsby-Smith, L., Gleeson, J. and Eyre, B.D., 2013. Groundwater‐derived dissolved inorganic and organic carbon exports from a mangrove tidal creek: The missing mangrove carbon sink?. Limnology and Oceanography, 58(2), pp.475-488.

Mergulhao, L.P., Guptha, M.V.S., Unger, D. and Murty, V.S.N., 2013. Seasonality and variability of coccolithophore fluxes in response to diverse oceanographic regimes in the Bay of Bengal: Sediment trap results. Palaeogeography, palaeoclimatology, palaeoecology, 371, pp.119-135.

Ovalle, A.R.C., Rezende, C.E., Lacerda, L.D. and Silva, C.A.R., 1990. Factors affecting the hydrochemistry of a mangrove tidal creek, Sepetiba Bay, Brazil. Estuarine, Coastal and Shelf Science, 31(5), pp.639-650. Sarma, V.V.S.S., 2003. Monthly variability in surface p CO2 and net air‐sea CO2 flux in the Arabian Sea. Journal of Geophysical Research: Oceans, 108(C8).

Sarma, V.V.S.S., Kumar, M.D. and George, M.D., 1998. The central and eastern Arabian Sea as a perennial source of atmospheric carbon dioxide. Tellus B: Chemical and Physical Meteorology, 50(2), pp.179-184. Stoll, H.M., Arevalos, A., Burke, A., Ziveri, P., Mortyn, G., Shimizu, N. and Unger, D., 2007. Seasonal cycles in biogenic production and export in Northern Bay of Bengal sediment traps. Deep Sea Research Part II: Topical Studies in Oceanography, 54(5-7), pp.558-580.

Yang, W.B., Yuan, C.S., Tong, C., Yang, P., Yang, L. and Huang, B.Q., 2017. Diurnal variation of CO2, CH4, and N2O emission fluxes continuously monitored in-situ in three environmental habitats in a subtropical estuarine wetland. Marine pollution bulletin, 119(1), pp.289-298.

---

## Author Comment (AC2) · 10 Jan 2020

Response to reviewer 2's comments:

Reviewer 2's comment:

The present study investigates water pCO2 at 8 different stations of mangrove surrounding waters (creek, island boundary, mid-river) at Dhanchi Island in the Sundarbans, India. The authors present an interesting high resolution data set (8 x 24h time-series, diurnal, tidal) at 1 min interval of pCO2 and find mangrove surrounding waters to be a weak source or sink of atmospheric CO2. The authors aim to reveal and identify why the here studied mangrove waters act as a net sink compared to previous studies, that are commonly found to be a source of CO2. They conclude that the reduced riverine input and increased buffering capacity from oceanic water is responsible for the low pCO2 in the mangrove waters. Although the data set is impressive and worthwhile publication, I am not convinced that the authors have sufficiently identified and discussed the low pCO2 at the different study locations based on their data. The discussion is very speculative (see reviewer#1, I mostly agree with reviewer #1: regarding the low TAlk of the marine end-member. This value is questionable. I also agree, that the Revelle cannot be used to explain the CO2 sink. I further agree with reviewer #1 that some of the data (optical, NEP, NEC) seem out of context and do not provide relevant information to explain the low pCO2.NEP and NEC calculations need to be included in the methods section. High salinity combined with high abundance of phytoplankton or benthic micro-algae could be an explanation for the low pCO2.)

Authors' Response:

Thank you very much Reviewer 2 for the constructive comments on the improvement of our manuscript.

(1) Low TA in MEM during the post-monsoon season:

We agree with Reviewer 1 and Reviewer 2 that TAlk value is unusually low in marine end member (MEM). That is why we have already explained the background of MEM in details in the methodology section (from Line 300 to 312) using published and our unpublished data. Briefly, the TA data of the present study are also in good agreement with the few available published work in this transition zone conducted during the post-monsoon season of 2010-2011 and 2011-2012 (January and February) (Akhand et al. 2012; 2013), showing low values during post monsoon season (the season of the present study). From the unpublished data, the TA of MEM was found to be higher during pre-monsoon (1932 $\mu$mol kg-1) and monsoon (1879 $\mu$mol kg-1), than post-monsoon (1646 $\mu$mol kg-1). Indeed, the sampling and cruise of Goyet et al. (1999) was conducted between 29 August 1995 and 16 October 1995, which was of different season and far offshore (approx. 10o N latitude). Furthermore, the present study used

the standard preservation method for TA samples and used the gran plot method with an automatic titrator (batch sample analyser). Moreover, a certified reference material was procured and used to maintain the company specific accuracy of the batch sample analyser. Therefore, we believe that the present work is reporting sufficient precision to be published.

To further examine the interesting phenomena of low TA in MEM during the post-monsoon season, we also examined the modelled data archive (https://esgf-node.llnl.gov/projects/esgf-llnl/) (Dunne et al. 2012). The spatial resolution of the data is coarse, i.e. 1o; hence, not exactly comparable with our field observation data because these data covers whole area of the coastline to the offshore transition zone, but we believe that it can be used as a supporting data to understand the carbonate chemistry of MEM. We used the spatial extent of 20o to 21oN and 87oto 91oE to extract the data which covers a substantial area of the northern Bay of Bengal and for the years 1995-1996 to 2004-2005 (10 consecutive years). The extracted data showed the mean TA value being 2.11 mmol kg-1(pre-monsoon, from February to May), 2.04 mmol kg-1(monsoon, from June to September) and 1.98 mmol kg-1(post-monsoon, from October to January), indicating the same pattern as our field observation data, i.e., TA never reaches 2.2 mmol kg-1 in the transition zone, and both TA and DIC decrease during post-monsoon season. The decrease in both TA and DIC during post-monsoon seasons might be because of phytoplanktonic calcifies (foraminifera and coccolithophores) blooming during post-monsoon season (Biswas et al. 2004) in the northern Bay of Bengal (Stoll et al. 2007; Mergulhao et al. 2013).

Overall, the phenomenon of low TA in MEM during the post-monsoon season is indeed a very interesting topic of research. However, addressing further this phenomenon is out of scope for our present work.

(2) Use of Revelle factor:

We did not focus on the 'CO2 sink character' in our manuscript, because, as a whole,

mangrove surrounding water of Sundarban (Indian part) is net source of CO2 covering all seasons throughout the year and considering upper to lower estuary. The fact is evidenced from number of previous studies and in parity with the upscaled data of the present study (section 4.5). The tittle of the present manuscript also represents CO2 'evasion', not 'sink'. We explained in the present version of the manuscript that the CO2 efflux / evasion rate of Sundarban is much lesser than the recently estimated world average due to high buffer capacity (low revelle factor). We also agree with Reviewer 1 and Reviewer 2, that biological uptake can also an important mechanism explaining for low pCO2. We did not state about biological uptake in the present manuscript because biological uptake seems to be minor. We showed using Fig. 5a that most of the $\Delta$pCO2 value was positive, which indicates the study area exhibited heterotrophy. We'll add sentence(s) for better clarification of this issue. Furthermore, we collected time series phytoplankton standing stock (chlorophyll-a fluorescence) data using a fluorometer but did not get any significant correlation with pCO2 (water). However, we agree with the reviewers' concern and will present the time series data of chlorophyll-a in the supplementary material (to be included in figure S1). We'll also add discussion about less correlation with pCO2 (water) and chlorophyll-a to show the less significant effect of phytoplankton productivity in pCO2(water) and CO2 flux, especially in the stations showing CO2 sink.

(3) Optical signature, NEP and NEC data:

According to the suggestion of Reviewer 1 and Reviewer 2, we'll eliminate NEP, NEC and optical signature, i.e. CDOM part from the whole manuscript.

Authors' changes in the manuscript: We'll add sentence(s) and citation(s) to describe the probable reason behind such low magnitudes of TAlk and DIC during post-monsoon in the marine end member in 'Data Analysis' (section 2.6) under 'Materials and methods' while describing the carbonate chemistry of the MEM.

We'll add sentence(s) for better understanding of the revelle factor issue. We'll add

a separate paragraph to clarify about the 'Biological factor' in relation with low pCO2. We'll add sentence(s) on the heterotrophic nature of the study area showing Fig. 5a, regarding ∆pCO2 value. We'll add chlorophyll-a time series data in the Fig. S1 and will add sentence(s) on the relation between pCO2 and Chlorophyll-a in the 'Discussion' section along with necessary sentence(s) in the 'Materials and methods' and 'Results' section. We'll add sentence(s) and references(s) on the special low pCO2 character of the Bay of Bengal in the 'Introduction' section.

We'll eliminate optical signatures (i.e. CDOM), NEP and NEC part from the whole manuscript.

Reviewer 2's comment:

The authors mention that there is no (or almost none) riverine connection. Yet, they use a freshwater end-member upstream to estimate the conservative mixing lines, which does not makes sense if there is no riverine connection. Similarly, the marine end member seems questionable with a salinity of 26, which is very close to the mangrove waters (salinity 25-26).

Authors' Response: Considering Reviewer 2's comments and rechecking previous studies (for example Ray et al. 2018; Dutta et al. 2019) in Sundarban (Indian part) on CO2/ carbon dynamics, we decided to eliminate all the wordings and sentences, which seems there is no riverine connection (or almost none), and replace the confusing phrases like "negligible riverine freshwater input" to "indirect and lesser riverine freshwater input". This is because the quantification of riverine freshwater input from Hooghly River to the Sundarban (Indian part) was not done previously using hydrological modelling and out of scope for the present study, It is well established fact, that Indian part of Sundarbans are not getting riverine freshwater directly from its upstream rivers (for example Matla, saptamukhi, Thakuran etc.), but there are waterways from where the riverine freshwater is entering to the Indian part of Sundarbans, as shown in high resolution maps (for example google earth). Namely, Hatania Doania canal

which connects Hooghly River with Saptamukhi Estuary (Sundarban) (Ray et al. 2018) through Muriganga River. Riverine freshwater from Hooghly River enters through this canal and then spreads to Sundarban by different waterways. We'll show the Hatania Doania canal, which connects Sundarban (Indian part) with Hooghly River, in the Fig.1 of the revised manuscript. Hence, we think "indirect riverine freshwater input lesser than other river dominated estuary" will be more appropriate phrase and will clarify the ambiguity. It is also a well-established fact by previous studies that more riverine freshwater input causes higher pCO2(water) and subsequent air-water CO2 efflux in comparison with lesser input (Jiang et al. 2008; Maher and Eyre 2012; Akhand et al. 2016). We'll add supporting reference(s) and add sentence(s) to 'Introduction' section to clarify the ambiguity and to describe the exact conditions.

The salinity data of the marine end member is comparable with previously published data (Akhand et al. 2012, 2013) in the same region during the same post-monsoon season. The salinity data of the marine end member during the same month (January and February) of sampling is also in good agreement with the data archive of salinity (https://www.nodc.noaa.gov/OC5/woa18/). We anticipate that the salinity at the site is mainly affected by the 'Bengal Fan'. As Hooghly-Bhramhaputra Rivers are huge source of freshwater to the northern Bay of Bengal and because of Bengal Fan, the salinity of 60 km off the coast was found 26.9 in the same month (reported in this manuscript), which is a unique feature in this region. As mainly such a saline water mass enters and recedes within Matla Estuary, the salinity range of 25-26 which observed in the study area during post-monsoon is quite natural.

Authors' changes in the manuscript:

All the ambiguous words, phrases and sentences, like "negligible riverine freshwater input" will be replaced by "indirect and lesser riverine freshwater input". We'll add supporting sentence(s) to the 'Introduction' section to clarify the ambiguity and to describe the exact conditions of both indirect riverine freshwater inputs and the characteristics of the marine end member, i.e. northern Bay of Bengal. We'll show the Hatania Doania canal, which is the main waterway of entering riverine freshwater from the Hooghly River to the Sundarban (Indian part), in the Fig.1 of the revised manuscript and edit the Fig.1 caption accordingly with the citation of Ray et al. (2018).

Reviewer 2's comment:

Secondly, the station C1 and C2 are substantially different and should not be treated as one group. To me, station C2 seems like the only "real" mangrove site. As in several other previously studied mangrove surrounding water locations cited in this manuscript, a single creek ending in a mangrove forest is the ideal location to study tidal and temporal variability and fluxes of inorganic carbon and dissolved gases (tidal pumping). C2 has no connection other than to the estuary. In contrast, C1 is not a "creek" but more a branch or tributary of the main estuary channel that connects the left and right (Thakun) estuary channels, therefore is influenced by biogeochemical processes of both channels. I disagree that 20 meter width is indicative of a "very narrow creek". I am not surprised to see the very low pCO2 in the main estuary (not river) channel and close by island boundary. These study sites (MR, IB, C1) seem more indicative of a marine environment with low change in salinity (salinity 24-27).

What is the effect from macro-tides compared to meso or micro-tides? Fig.2 MR3 shows a typical diurnal trend of CO2 rather than a tidal trend. The authors identified correctly that the term "mangrove surrounding waters" can be ambiguous. StationMR1-3 might be better to compare to (previous) estuary CO2 emissions than mangrove CO2 emissions?

The latest global mangrove forest distribution is Bunting et al. (2018) Bunting P, Rosenqvist A, Lucas RM, Rebelo LM, Hilarides L, Thomas N, Hardy A, Itoh T, Shimada M,Finlayson CM. 2018. The global mangrove watch - a new 2010 global baseline of mangrove extent. Remote Sensing 10 (10) DOI: 10.3390/rs10101669

Authors' Response: We agree with the concern of Reviewer 2, that categorisation of Creek, Island Boundary and Main River are somewhat ambiguous. Also, thanks to

Reviewer 2, we found that the location of sampling station C1 was wrongly placed in the Fig.1. We'll rectify this unintentional error in our revised manuscript. However, the C1 is also a dead-end creek with a width of 20 m and becoming narrower with more inside, which is very common in Sundarbans. Therefore, we'll delete the word 'narrow', before 'creek'. We'll only categorise our sampling stations as Creek (C1 and C2) and River (R1 to R6) in the revised manuscript. We anticipate that, the less variability of salinity is due to the dominance by the marine water of the Bay of Bengal. Namely, the salinity of the northern Bay of Bengal is much lesser than the open ocean as northern Bay of Bengal receives huge riverine fresh water from Hooghly and Bramhaputra Rivers and because of 'Bengal Fan'. Moreover, freshwater input from the upstream, especially during study period (dry season i.e. post-monsoon), is lesser as discussed in the response of the previous comments.

In turn, we think that IB and MR stations (R stations in the revised manuscript) should also be considered as 'mangrove surrounding water'. The contribution of mangrove derived organic matter is higher in the POM of the creek stations than that of the IB and MR stations, but there are some contributions of mangrove derived organic matter in IB and MR stations (will be categorised as R stations in the revised manuscript) as evidenced from three end-member mixing model (L603-616 of the present version of the manuscript; Fig. S2 and Tables S3 and S4 of supplementary material). Moreover, IB and MR stations (R stations in the revised manuscript) are the only possible way to supply of mangrove derived dissolved and particulate carbon to the coastal sea i.e. Bay of Bengal.

Nevertheless, as we wrote in the 'Introduction' section, there is no well-established demarcation of the term 'mangrove surrounding water', and it might be misleading to demark it with certainty. Similarly, no ideal definition of the 'mangrove surrounding water' is possible. Only way is to consider the waters situated in the vicinity of mangrove forests / stretch / patches. Previously, researchers used the term 'mangrove surrounding water', 'mangrove water' or 'mangrove waterway' alternatively for creek, river or

marine water situated in the vicinity of mangroves (for example, Borges et al. 2003; Bouillon et al. 2007; Sippo et al. 2016; Rosentreter et al. 2018; in Sundarban: Biswas et al. 2004). We'll add sentence(s) for better clarification of this point in 'Introduction' section.

Regarding the query 'What is the effect from macro-tides compared to meso or micro-tides?', we just wanted to emphasize on the quick rate of dilution, we used this term in the manuscript. We anticipate that the quick dilution is facilitated by the meso to macro-tidal nature of the study cite as stated in L532 ("large tidal amplitude") of the present version of the manuscript. We'll add the phrase 'meso to marco-tidal estuary' in this line for better clarification.

Authors' changes in the manuscript: We'll categorise our sampling stations as Creek (C1 and C2) and River (R1 to R6), instead of Creek (C1 and C2), Island Boundary (IB1, IB2 and IB3) and Main River (MR1, MR2 and MR3). We'll delete the word 'narrow' before the word 'creek'. We'll add sentence(s) to clarify about the term 'mangrove surrounding water' in the 'Introduction' section. We'll add the phrase 'meso to marco-tidal estuary' in L532 for better clarification.

Reviewer 2's comment: L97-98 what is the difference between "mangrove surrounding waters" and "mangrove waters" in this context here? I would suggest to define what you mean with "mangrove surrounding waters" at the beginning of the manuscript and then use this term consistently throughout the manuscript.

Authors' Response: Same to our response to the previous comment, we want to convey, that 'to define' 'mangrove surrounding water', might be misleading. However, we'll add sentence(s) to clarify about the term 'mangrove surrounding water' in the 'Introduction' section. We'll change the sentence (L97-98) and will use the term 'mangrove surrounding water' throughout the manuscript.

Authors' changes in the manuscript: We'll add sentence(s) to clarify about the term 'mangrove surrounding water' in the 'Introduction' section. We'll change the sen-
tence (L97-98) and will use the term 'mangrove surrounding water' throughout the manuscript.

Reviewer 2's comment: I would suggest to change the title. The term "evasion rate" implies an efflux of CO2 from water to the atmosphere while the authors aim to highlight the influx. Alternatively title similar to this: "Low pCO2 in mangrove surrounding water in the Sundarbans".

Authors' Response: We believe that the title is leaved unchanged as our study did not highlight "CO2 influx", because, as a whole (throughout the year and including upper to lower part of the estuary), mangrove surrounding water of Sundarban (Indian part) acts as a source of CO2. This fact is well established from the previous works and from the upscaled data of the present study (sees section 4.5). We wanted to convey, that the CO2 efflux rate (evasion rate) is much lesser than the recently estimated world average.

Authors' changes in the manuscript: No change.

Reviewer 2's comment: The gas transfer velocity is the highest uncertainty in the gas flux computation; therefore k parameterisations should be chosen carefully. It is advisable to compare fluxes based on several different k parameterisations (not just one) in dynamic tidal ecosystems such as mangrove estuaries. It would be interesting to see how much this would change the average influx/efflux.

Authors' Response: We fully agree with Reviewer 2 at this point. We'll calculate the flux using different k parameterisation and add sentence(s) regarding that calculation.

Authors' changes in the manuscript: We'll calculate the flux using different k parameterisation and add sentence(s) regarding that calculation.

Reviewer 2's comment: L498 "pCO2 concentration" is wrong. It is pCO2 or CO2 concentration (e.g. $\mu$M).

Authors' Response: We agree and will change accordingly.

Authors' changes in the manuscript: The word 'concentration' will be deleted in L498.

Reviewer 2's comment: L521-529: This is unclear. Do the authors suggest that the source of DIC is a mix of all the possible sources listed in this paragraph?

Authors' Response: Probably, Reviewer 2 indicated about the paragraph of L508-520. L521-529 is supporting paragraph for the previous one. Yes, this part (L508-520) is a scientific assumption of the mixing of possible sources. We'll add sentence(s) for better clarification.

Authors' changes in the manuscript: We'll add sentence(s) for better clarification in the revised manuscript.

Reviewer 2's comment: L436-439, L601 The authors suggest "rapid transport to the coastal ocean". Do they mean rapid flushing of pore water? Or tidal pumping? Why rapid dilution? This is unclear. It might be helpful to calculate the freshwater flushing times for the estuary to support this hypothesis. Although with no or very little riverine input I assume very low flushing.

Authors' Response: We agree with Reviewer 2, that the word 'rapid' is ambiguous in L436-439. We'll delete the word 'rapid' from L436-439. Ray et al. (2018a) extensively studied the lateral transport of mangrove derived carbon in the adjacent coastal sea, i.e. Bay of Bengal. They stated the reason of "rapid transport to the coastal ocean" as shorter residence time due to large tidal amplitudes and estuarine geometry ("funnelling effect"). We cited their work (L530-535) and references therein to support our hypothesis. These reasons include both 'tidal flushing' and 'tidal pumping'. These reasons have been discussed in L530-535. The present study has not dealt with lateral flux and it is out of ambit to calculate 'freshwater flushing times', because of the unavailability of the residence time and other necessary data in such micro level.

We think, the term 'rapid transport' is appropriate in L601, because explained well previously in L530-535.

Authors' changes in the manuscript: We'll delete the word 'rapid' from L436-439. We'll replace the citation "Ray et al. 2018a" by "Ray et al. 2018a and the references therein". The word 'rapid' remained unchanged in L601.

Reviewer 2's comment: Yes, the term "pCO2-lean seawater" is awkward.

Authors' Response: We'll replace the term 'pCO2-lean' by 'low pCO2' throughout the manuscript.

Authors' changes in the manuscript: We'll replace the term 'pCO2-lean' by 'low pCO2' throughout the manuscript.

References

Akhand, A., Chanda, A., Dutta, S. and Hazra, S., 2012. Air–water carbon dioxide exchange dynamics along the outer estuarine transition zone of Sundarban, northern Bay of Bengal, India. Indian Journal of Geo-Marine Science,41(2), pp.111-116.

Akhand, A., Chanda, A., Dutta, S., Manna, S., Hazra, S., Mitra, D., Rao, K.H. and Dadhwal, V.K., 2013. Characterizing air–sea CO2 exchange dynamics during winter in the coastal water off the Hugli-Matla estuarine system in the northern Bay of Bengal, India. Journal of oceanography, 69(6), pp.687-697.

Akhand, A., Chanda, A., Manna, S., Das, S., Hazra, S., Roy, R., Choudhury, S.B., Rao, K.H., Dadhwal, V.K., Chakraborty, K. and Mostofa, K.M.G., 2016. A comparison of CO2 dynamics and air‐water fluxes in a river‐dominated estuary and a mangrove‐dominated marine estuary. Geophysical Research Letters, 43(22), pp.11-726.

Biswas, H., Mukhopadhyay, S.K., De, T.K., Sen, S. and Jana, T.K., 2004. Biogenic controls on the air—water carbon dioxide exchange in the Sundarban mangrove environment, northeast coast of Bay of Bengal, India. Limnology and Oceanography, 49(1), pp.95-101.

Borges, A.V., Djenidi, S., Lacroix, G., Théate, J., Delille, B. and Frankignoulle, M., 2003. Atmospheric CO2 flux from mangrove surrounding waters. Geophysical Research Letters, 30(11).

Bouillon, S., Dehairs, F., Velimirov, B., Abril, G. and Borges, A.V., 2007. Dynamics of organic and inorganic carbon across contiguous mangrove and seagrass systems (Gazi Bay, Kenya). Journal of Geophysical Research: Biogeosciences, 112(G2).

Dunne, J.P., John, J.G., Adcroft, A.J., Griffies, S.M., Hallberg, R.W., Shevliakova, E., Stouffer, R.J., Cooke, W., Dunne, K.A., Harrison, M.J. and Krasting, J.P., 2012. GFDL's ESM2 global coupled climate–carbon earth system models. Part I: Physical formulation and baseline simulation characteristics. Journal of Climate, 25(19), pp.6646-6665.

Dutta, M.K., Kumar, S., Mukherjee, R., Sanyal, P. and Mukhopadhyay, S.K., 2019. The post-monsoon carbon biogeochemistry of the Hooghly–Sundarbans estuarine system under different levels of anthropogenic impacts. Biogeosciences, 16(2), pp.289-307.

Goyet, C., Coatanoan, C., Eischeid, G., Amaoka, T., Okuda, K., Healy, R. and Tsunogai, S., 1999. Spatial variation of total CO2 and total alkalinity in the northern Indian Ocean: A novel approach for the quantification of anthropogenic CO2 in seawater. Journal of Marine Research, 57(1), pp.135-163.

Jiang, L.Q., Cai, W.J. and Wang, Y., 2008. A comparative study of carbon dioxide degassing in river‐and marine‐dominated estuaries. Limnology and Oceanography, 53(6), pp.2603-2615.

Maher, D.T. and Eyre, B.D., 2012. Carbon budgets for three autotrophic Australian estuaries: Implications for global estimates of the coastal air‐water CO2 flux. Global Biogeochemical Cycles, 26(1).

Mergulhao, L.P., Guptha, M.V.S., Unger, D. and Murty, V.S.N., 2013. Seasonality and variability of coccolithophore fluxes in response to diverse oceanographic regimes in the Bay of Bengal: Sediment trap results. Palaeogeography, palaeoclimatology,

palaeoecology, 371, pp.119-135.

Ray, R., Baum, A., Rixen, T., Gleixner, G. and Jana, T.K., 2018. Exportation of dissolved (inorganic and organic) and particulate carbon from mangroves and its implication to the carbon budget in the Indian Sundarbans. Science of the Total Environment, 621, pp.535-547.

Rosentreter, J.A., Maher, D.T., Erler, D.V., Murray, R. and Eyre, B.D., 2018. Seasonal and temporal CO2 dynamics in three tropical mangrove creeks–A revision of global mangrove CO2 emissions. Geochimica et Cosmochimica Acta, 222, pp.729-745.

Sippo, J.Z., Maher, D.T., Tait, D.R., Holloway, C. and Santos, I.R., 2016. Are mangroves drivers or buffers of coastal acidification? Insights from alkalinity and dissolved inorganic carbon export estimates across a latitudinal transect. Global Biogeochemical Cycles, 30(5), pp.753-766.

Stoll, H.M., Arevalos, A., Burke, A., Ziveri, P., Mortyn, G., Shimizu, N. and Unger, D., 2007. Seasonal cycles in biogenic production and export in Northern Bay of Bengal sediment traps. Deep Sea Research Part II: Topical Studies in Oceanography, 54(5-7), pp.558-580.